https://doi.org/10.1038/s41467-022-31358-5　　**OPEN**

# Limited effects of m$^6$A modification on mRNA partitioning into stress granules

Anthony Khong [1,2], Tyler Matheny[1,3], Thao Ngoc Huynh[1], Vincent Babl[1] & Roy Parker [1,2] ✉

The presence of the m$^6$A modification in mammalian mRNAs is proposed to promote mRNA recruitment to stress granules through the interaction with YTHDF proteins. We test this possibility by examining the accumulation of mRNAs in stress granules in both WT and Δ*METTL3* mES cells, which are deficient in m$^6$A modification. A critical observation is that all m$^6$A modified mRNAs partition similarly into stress granules in both wild-type and m$^6$A-deficient cells by single-molecule FISH. Moreover, multiple linear regression analysis indicates m$^6$A modification explains only 6% of the variance in stress granule localization when controlled for length. Finally, the artificial tethering of 25 YTHDF proteins on reporter mRNAs leads to only a modest increase in mRNA partitioning to stress granules. Since most mammalian mRNAs have 4 or fewer m$^6$A sites, and those sites are not fully modified, this argues m$^6$A modifications are unlikely to play a significant role in recruiting mRNAs to stress granules. Taken together, these observations argue that m$^6$A modifications play a minimal, if any, role in mRNA partitioning into stress granules.

[1] Department of Biochemistry, University of Colorado, Boulder, CO 80309, USA. [2] Howard Hughes Medical Institute, Chevy Chase, MD 20815, USA. [3] RNA Bioscience Initiative, University of Colorado Anschutz Medical Campus, Aurora, CO 80045, USA. ✉email: roy.parker@colorado.edu

Stress granules are cytoplasmic molecular condensates composed of non-translating messenger ribonucleoproteins (mRNPs). Stress granules form when there is an increase in the pool of non-translating mRNPs, which often occurs when cells undergo a variety of stress conditions including oxidative stress, hypoxia, and heat shock that downregulate translation initiation[1,2]. Stress granules are of interest since they are thought to play roles in a variety of diseases such as viral infection, cancer, and neurodegenerative disorders. Moreover, their investigation may provide insights into other RNA and protein condensates such as the nucleolus, P-bodies, and germ granules[1,3–5].

Recent studies have elucidated the composition of stress granules in a variety of stress conditions[6–11]. One of the major conclusions from these studies is the mRNAs that are preferentially enriched in stress granules are biased towards poorer translation efficiency and longer length[9,10]. Despite this length bias, mRNAs of the same length can show different recruitment into stress granules arguing that there can also be sequence-specific information that affects mRNPs partitioning into stress granules.

Length-independent effects of stress granule recruitment have been argued to occur by the m[6]A modification increasing mRNA partitioning into stress granules through the binding of YTHDF proteins[12,13]. However, this possible mechanism has not been directly tested by inhibiting m[6]A modification and examining the effect on mRNP partitioning into stress granules. Herein, we compared the localization of poly-m[6]A mRNAs in wildtype and ΔMETTL3 mES cells in stress granules and discovered the poly-m[6]A mRNAs are enriched similarly in stress granules in both cell types. These results suggest m[6]A plays little to no role in recruiting endogenous mRNA to stress granules. Moreover, we observed tethering up to 25 YTHDF proteins to a reporter mRNA has only a modest increase in the reporter mRNA partitioning into stress granules. Therefore, in an artificial reporter system, YTHDF proteins can recruit RNAs to stress granules, but this requires significantly many more proteins than what is normally observed for the number of m[6]As that are seen for endogenous mRNAs. This argues m[6]A modification plays little role in the recruitment of endogenous mRNAs to stress granules.

## Results

### Observations that argue m[6]A targets mRNAs to stress granules.

In previous work, three main observations were used to argue that m[6]A modification targets mRNAs to stress granules by providing binding sites for YTHDF proteins, whose intrinsically disordered regions (IDRs) then promote mRNPs entering stress granules[13]. First, YTHDF proteins were shown to undergo self-assembly in vitro through liquid-liquid phase separation (LLPS), which is dependent on their IDRs, and increased by m[6]A modified RNAs. However, whether this in vitro LLPS is relevant to the cell was not tested. This is an issue since many proteins can undergo LLPS in vitro, particularly at high concentrations and in the absence of competing proteins[14].

A second observation used to argue that m[6]A modification promotes mRNAs entering RNP granules is the demonstration that the YTHDF2 protein localizes to P-bodies in the absence of stress and to stress granules during stress. However, the partitioning of YTHDF2 is dependent on RNA binding since YTHDF2 proteins fail to associate with stress granules or P-bodies when RNA binding is blocked by deletion of the m[6]A methylase (ΔMETTL14), or by expressing an RNA-binding mutant YTHDF2 protein in cells[13]. This argues that YTHDF2 partitioning into RNP granules is due to binding to RNA and indicates that the YTHDF2 IDR is not sufficient for stress granule partitioning. The simplest interpretation of this observation is

that when YTHDF proteins are bound to mRNAs, they can be carried into mRNP granules such as P-bodies and stress granules.

The third argument presented that m[6]A promotes the recruitment of poly-m[6]A mRNA to stress granules relies on single-molecule FISH data. Specifically, it is reported that two poly-m[6]A modified mRNAs, Fem1b and Fignl1, are recruited to stress granules more effectively (~60%) than two non-methylated m[6]A mRNAs, Grk6 and Polr2a, to stress granules (~30%). Since these mRNAs are of similar size, which is relevant since the length can affect mRNAs partitioning into stress granules[9], it was inferred that their difference in stress granule recruitment was due to differential m[6]A methylation.

### mRNA partitioning to stress granule unchanged by m[6]A loss.

In principle, the differential localization of the Fem1b, Fignl1, Grk6, and Polr2a mRNAs could be due to m[6]A differences or to other features of each mRNA. We analyzed the partitioning of several poly m[6]A mRNAs to stress granules in wildtype and ΔMETTL3 mES cells, a cell line created by Batista et al.[15]. The ΔMETTL3 removes the m[6]A mRNA writer[16], and its knockout reduces m[6]A levels on mRNAs globally in mES cells (Supplementary Fig. 1A). We inspected 10 poly-m[6]A modified mRNAs including Rictor, Zfp945, Aff1, Mtf2, Bahcc1, Mthfr, Dhx30, Tnrc6c, Lrp5, Zeb1, and one mono-m[6]A modified mRNA (Fig. 1, Supplementary Fig. 2A, B, Supplementary Movies 1–22). These 11 mRNAs all have significantly reduced m[6]A levels in the ΔMETTL3 cells as assessed by m[6]A immunoprecipitation followed by m[6]A mapping (Supplementary Fig. 1B, Batista et al.) and by qRT-PCR (Supplementary Fig. 1C). Nanog is a positive control that was previously shown by Batista et al. to be m[6]A depleted by m[6]A mapping and qRT-PCR (Supplementary Fig. 1C), which we reconfirmed with our analysis (Supplementary Fig. 1C). In addition, consistent with the role of m[6]A in reducing mRNA stability[17], we see a substantial increase in m[6]A modified mRNAs in ΔMETTL3 mES cells compared to wild-type mES cells (Supplementary Fig. 1D). This was also observed for Nanog mRNAs by Batista et al. (Supplementary Fig. 1D). Ten of these 11 mRNAs have multiple m[6]A mapped sites distributed in different mRNA regions (Supplementary Fig. 1B and Batista et al., which would be expected to increase any possible contribution from m[6]A modification.

A critical observation is that we observed no significant differences between WT and ΔMETTL3 mES cells in the stress granule accumulation of all 11 m[6]A mRNAs tested (Fig. 1, Supplementary Fig. 2A–B, Supplementary Movies 1–22). These results argue that m[6]A modification does not play a major role in the targeting of these mRNAs to stress granules.

We also looked at 3 other RNAs where the modification did not change between WT and ΔMETTL3 cells or was unmodified to begin with (Fem1b, Fignl1, and Polr2A) (Supplementary Fig. 1C, Supplementary Fig. 2C, D)). We see no changes in the localization of these mRNAs, and Fem1b and Fignl1 mRNA levels did not change as drastically between ΔMETTL3 and WT cells compared to the other m[6]A modified RNAs (Supplementary Fig. 1D).

### Length, not m[6]A, correlates with stress granule enrichment.

These results demonstrate that m[6]A modifications in mRNAs are not a strong contributor to mRNA partitioning in stress granules. However, prior work has described a correlation between the number of m[6]A sites on mRNAs[13,18] and enrichment in stress granules[9] or mRNP granules[10], respectively. However, this correlation may be fortuitous since the number of m[6]A sites and stress granule enrichment are both correlated with mRNA length (Fig. 2A)[9].

Poly-m$^6$A mRNAs

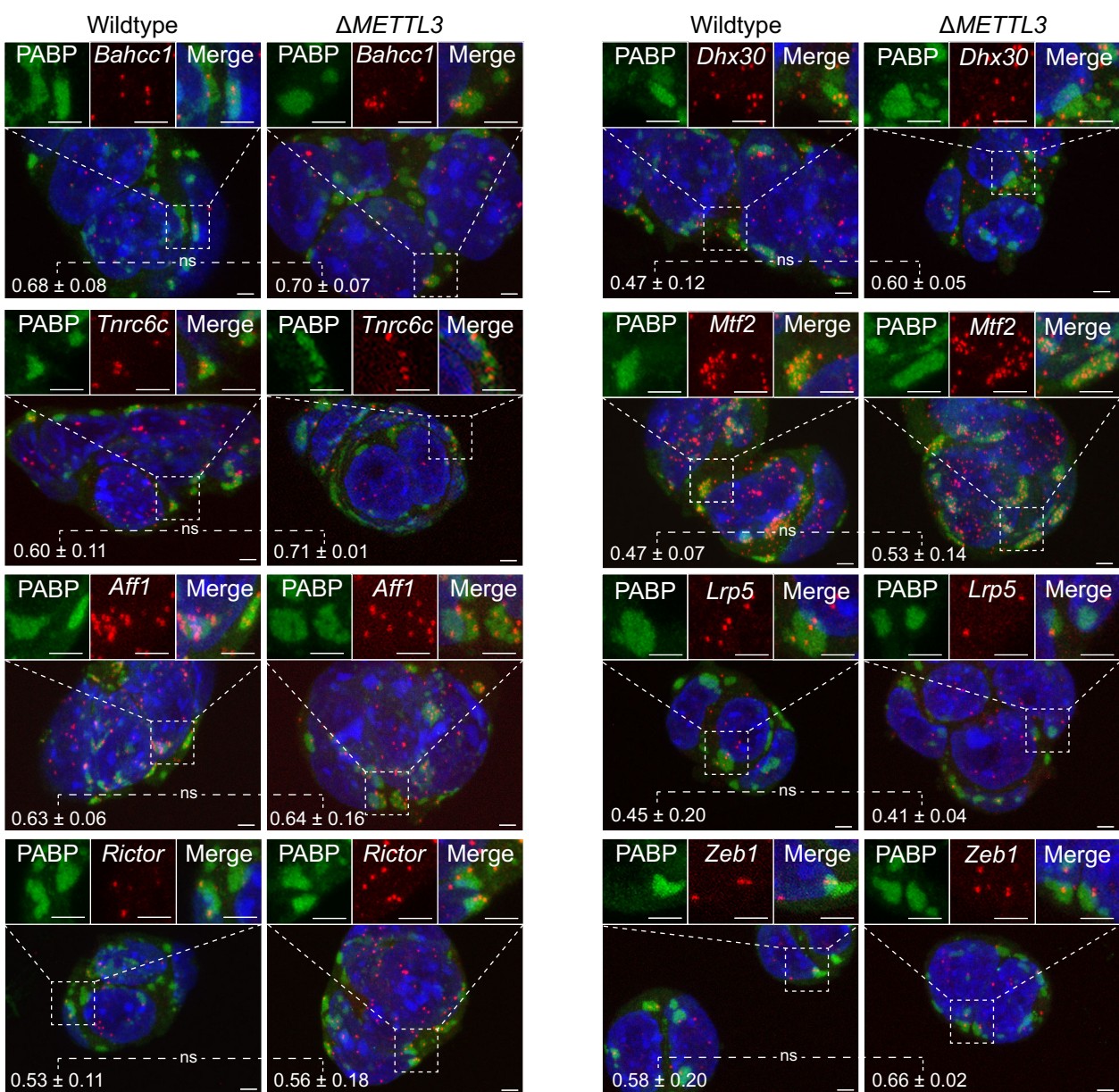

**Fig. 1 There is no difference in the fraction of polymethylated m$^6$A mRNAs in arsenite-induced stress granules between wildtype and Δ*METTL3* mES cells.** Representative images of wildtype and Δ*METTL3* mES cells stressed for 1 h with arsenite and costained with single-molecule FISH probes against *Bahcc1, Dhx30, Tnrc6c, Mtf2, Aff1, Lrp5, Rictor, and Zeb1* (red) and antibody against PABP protein (green). The nuclei are stained with DAPI (blue). Scale bar is 1 μM. Standard deviations are derived from three biological replicates. ns denotes not significant respectively (unpaired two-tailed Student's *t* test, $P > 0.05$). Source data are provided for this Figure (see data availability).

To computationally examine if m$^6$A modification contributes to mRNA partitioning into stress granules independent of length, we performed multiple regression analyses where we compared the effect of mRNA length on stress granule partitioning with or without an additional contribution of m$^6$A modification. We found that a linear regression model based on mRNA length alone showed an $R^2$ score of 0.41 for predicted stress granule enrichment vs. observed stress granule enrichment (Fig. 2B). A linear regression model based on the mapped m$^6$A sites per transcript and m$^6$A ratio[19] showed an $R^2$ score of 0.23 for predicted stress granule enrichment vs. observed stress granule enrichment (Fig. 2C).

While mRNA length was a better predictor of stress granule enrichment than m$^6$A modification, this does not rule out the possibility that m$^6$A plays an additional role in stress granule enrichment in concert with length. To parse the relative contributions of length and m$^6$A to stress granule enrichment, we built a multiple linear regression model using length, the mapped m$^6$A sites per transcript, and the m$^6$A ratio as predictors of stress granule enrichment (Fig. 2D). When we considered the combination of m$^6$A mapped m$^6$A sites per transcript and m$^6$A ratio and RNA length, our model improved from an $R^2$ value of 0.41 to 0.47, suggesting that m$^6$A modification could explain a maximum of an additional 6% of the variance in stress granule enrichment (Fig. 2D). We see similar results when we considered the individual contribution of m$^6$A ratio or m$^6$A mapped sites, which increased the $R^2$ value from 0.41 to 0.44 or 0.45,

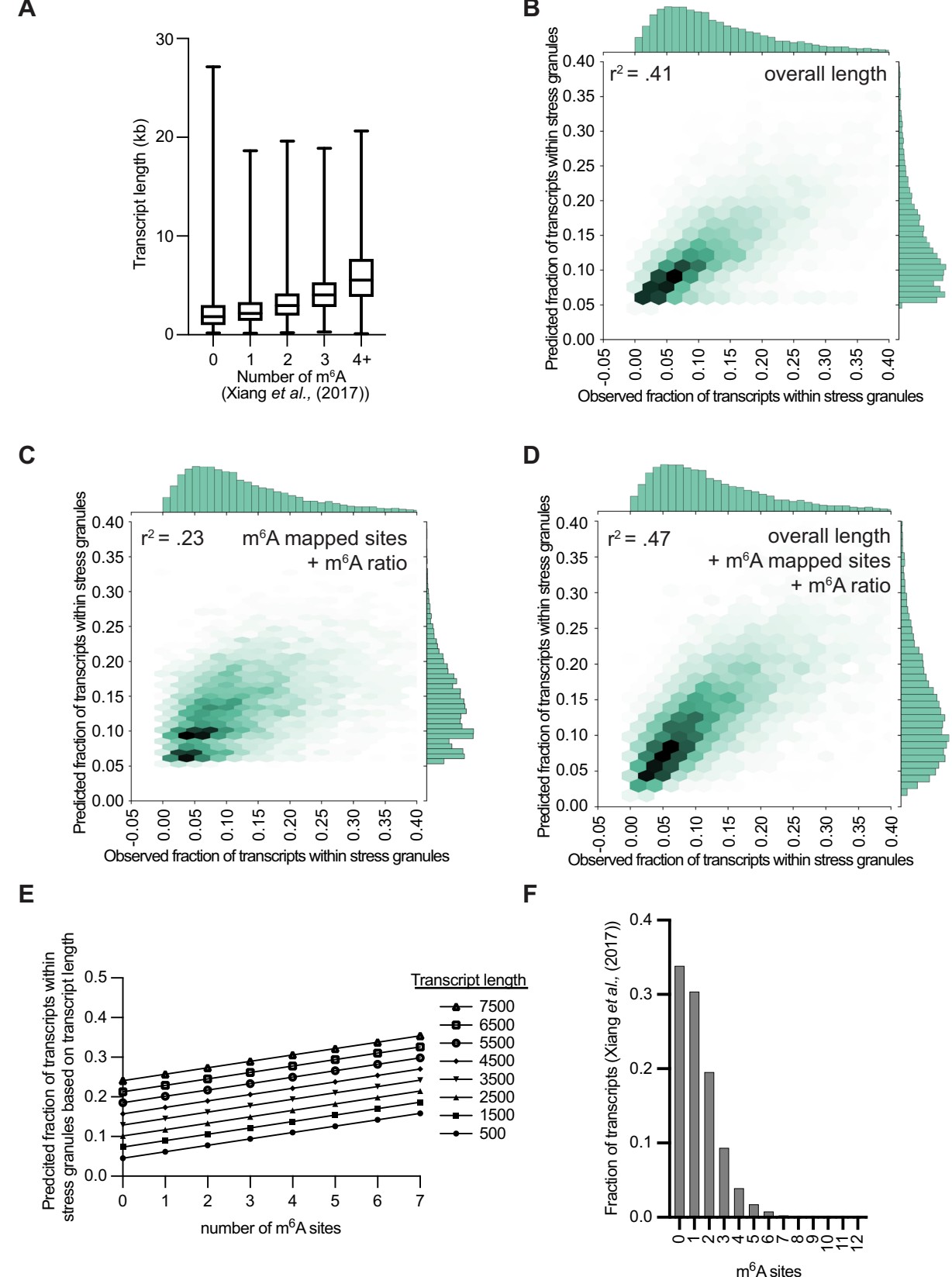

respectively (Supplementary Fig. 3A–C). These results suggest that m6A modification provides a minimal impact on mRNA partitioning into stress granules.

While adding m6A modifications improved our $R^2$ metric from 0.41 to 0.47, the $R^2$ metric does not give any insight into to

what degree m6A might enhance stress granule enrichment. To examine the degree to which transcript length and the number of m6A sites contribute to stress granule enrichment, we simulated data representing transcripts of lengths up to 7.5 kb and ranging from 0 to 7 m6A sites and analyzed the predicted

**Fig. 2 Bioinformatic prediction of m⁶A contribution to stress granule enrichment. A** Box plot of transcript length versus the number of mapped m⁶A sites in U-2 OS cells (Xiang et al.). Number of transcripts in each category are 3518, 3159, 2033, 970, and 712 for 0, 1, 2, 3, and 4 + m⁶A sites. The box plots are defined by minima, 25% percentile, median, 75% percentile, and maximum. For 0 m⁶A sites, the lengths are 192, 984, 1847, 3015, and 27125 for minima, 25% percentile, median, 75% percentile, and maximum, respectively. For 1 m⁶A sites, the lengths are 160, 1407, 2171, 3313, and 18626 for minima, 25% percentile, median, 75% percentile, and maximum, respectively. For 2 m⁶A sites, the lengths are 210, 1928, 2958, 4182, and 19607 for minima, 25% percentile, median, 75% percentile, and maximum, respectively. For 3 m⁶A sites, the lengths are 261, 2785, 4002, 5290, and 18865 for minima, 25% percentile, median, 75% percentile, and maximum, respectively. Finally, for 4 + m⁶A sites, the lengths are 110, 3848, 5446, 7685, and 20635 for minima, 25% percentile, median, 75% percentile, and maximum, respectively. **B** Scatterplot depicting predicted vs observed fraction of transcripts within stress granules. Predicted values were obtained from the linear regression model based on overall RNA length alone. **C** Same as B, but using a linear model based on the number of m⁶A sites per transcript and m⁶A ratio. **D** Same as B, but using a linear model based on both overall length, m⁶A sites per transcript, and m⁶A ratio. **E** Scatterplot depicting predicted fraction of transcripts within stress granules vs the number of m⁶A sites for RNA lengths from 500 bases to 7500 bases. **F** Histogram depicting the fraction of transcripts containing various m⁶A sites from Xiang et al. in U-2 OS cells. Source and code data are provided for this Figure (see data availability and code availability).

stress granule enrichment values of the multiple linear regression model (Fig. 2E). We observed that transcript length showed a stronger influence on stress granule enrichment than m⁶A modifications (Fig. 2E), which can be visualized by the increase in stress granule enrichment along the y-axis. The number of m⁶A modifications did show some effect on stress granule enrichment (x-axis), and we note that as the number of m⁶A sites per transcript increases (y-axis), stress granule enrichment also is predicted to slightly increase.

It should be noted that this simulation is likely over-estimating the contributions of m⁶A modifications for two reasons. First, we simulated transcripts containing up to 7 m⁶A sites, but >97% of transcripts are thought to have 4 or fewer m⁶A sites (Fig. 2F)[18]. Second, in this analysis using the number of m⁶A sites per transcript, we assumed that each m⁶A site is 100% modified, which is unlikely to be true since often specific m⁶A modification sites are modified at lower rates[20]. In this modeling, we found that each additional m⁶A modification on an RNA of a fixed 2.5 kb length would lead to a ~1.6% increase in stress granule enrichment (Fig. 2E). Thus, for the majority (>97%) of transcripts, which contain four or fewer m⁶A sites, we would predict m⁶A to maximally account for a 6.4% increase in enrichment.

Taken together, our computational analysis suggests there is a correlation between m⁶A modification and stress granule enrichment that could explain ~6% of the variance in stress granule partitioning. Our multiple linear regression model is limited by the fact that we compare m⁶A and stress granule enrichment from three different studies and different cell types[9,18,19]. More importantly, it is possible that this is simply a correlation due to other confounding variables. For example, mRNAs with higher amounts of exposed ssRNA might be more prone to assemble into stress granules[21,22], and might also have more accessible sites for m⁶A modification.

**YTHDF proteins minimally recruit RNAs to stress granules.** To experimentally examine if YTHDF proteins could have any impact on the recruitment of mRNAs to stress granules, we tethered 25 YTHDF proteins on *luciferase* reporter RNAs and examined its localization to stress granules using the λN-BoxB system[23]. By tethering 25 proteins to a single mRNA, which is an extreme condition, this assay should reveal if YTHDF proteins can have any effect on mRNAs partitioning into stress granules. *YTHDF1* and *YTHDF2* fused to GFP-λN transgene were transfected into U-2 OS cells stably expressing a *luciferase* reporter containing 25-BoxB stem-loops (Fig. 3A). We observed the tethering of YTHDF1-GFP-λN and YTHDF2-GFP-λN is functional because their expression reduced *25-BoxB-luciferase* reporter levels, but not *0-BoxB-luciferase* reporter levels (Fig. 3B), which is consistent with these YTHDF proteins enhancing mRNA degradation when associated

with mRNAs[17]. We quantified the localization of the reporter mRNA to stress granules by single-molecule FISH after arsenite stress for 60 minutes.

We observed that tethering YTHDF1 or YTHDF2 proteins could increase the recruitment of the *25-BoxB-luciferase* reporter mRNA to stress granules from an average of 11% to an average of 21 or 18% for YTHDF1-GFP-λN and YTHDF2-GFP-λN, respectively (Fig. 3C, D). This effect required the tethering of the YTHDF proteins to the reporter mRNA since no significant difference in stress granule recruitment was observed with the *0-BoxB-luciferase* reporter control mRNA (Fig. 3C, D). In side-by-side experiments with the same cell line, the level of recruitment by YTHDF proteins was similar to the effect of tethering G3BP-GFP-λN (18%). Therefore, these experimental results support the idea that the recruitment of YTHDF proteins to mRNA can increase their partitioning into stress granules. However, it is important to note that the number of tethered YTHDF proteins is far more than the number of m⁶A sites observed for endogenous mRNAs (Fig. 2F), arguing that individual m⁶A modifications play at most a minor role (<~6%) in the recruitment of endogenous mRNAs to stress granules (Fig. 2).

The YTHDF tethering assays should be interpreted with caution for a number of issues that may overemphasize the role of m⁶A recruiting mRNAs to stress granules. First, the Kd for BoxB elements binding to λN proteins is in the order of ~1.3 nM[24], which is approximately 500-fold greater than the interaction between YTHDF2 proteins and m⁶A modified mRNAs (~2.54 μM)[25]. Second, BoxB-containing RNAs do not have to compete with other mRNAs for binding to λN-fusion proteins in the tethering assay. In contrast, m6A-containing RNAs are likely competing with other m⁶A RNAs for binding to YTHDF proteins. Thus, this assay has many significant drawbacks that may artificially enhance the impact of m⁶A on mRNA recruitment to stress granules.

Several observations suggest that YTHDF and G3BP proteins behave similarly and share apparently contradictory properties. One shared property of YTHDF and G3BP proteins is that they can both affect the formation of stress granules when assessed by the overall area of stress granules[26–28]. Despite this role in stress granule formation, neither the interaction of either G3BP proteins (based on CRISPR knockout and stress granule transcriptome analysis[23]) nor YTHDF proteins (as assessed by the absence of m⁶A modification), affect the partitioning of mRNAs into stress granules. We suggest the resolution of this apparent conflict is due to the difference between examining the partitioning of an individual mRNA into stress granules compared to examining the formation of stress granules in bulk. The targeting of any individual mRNA to a stress granule is a summation of many interactions and therefore, any individual interaction makes little or no difference[23]. In contrast, the overall assembly of a stress granule is a highly cooperative process, and if the average

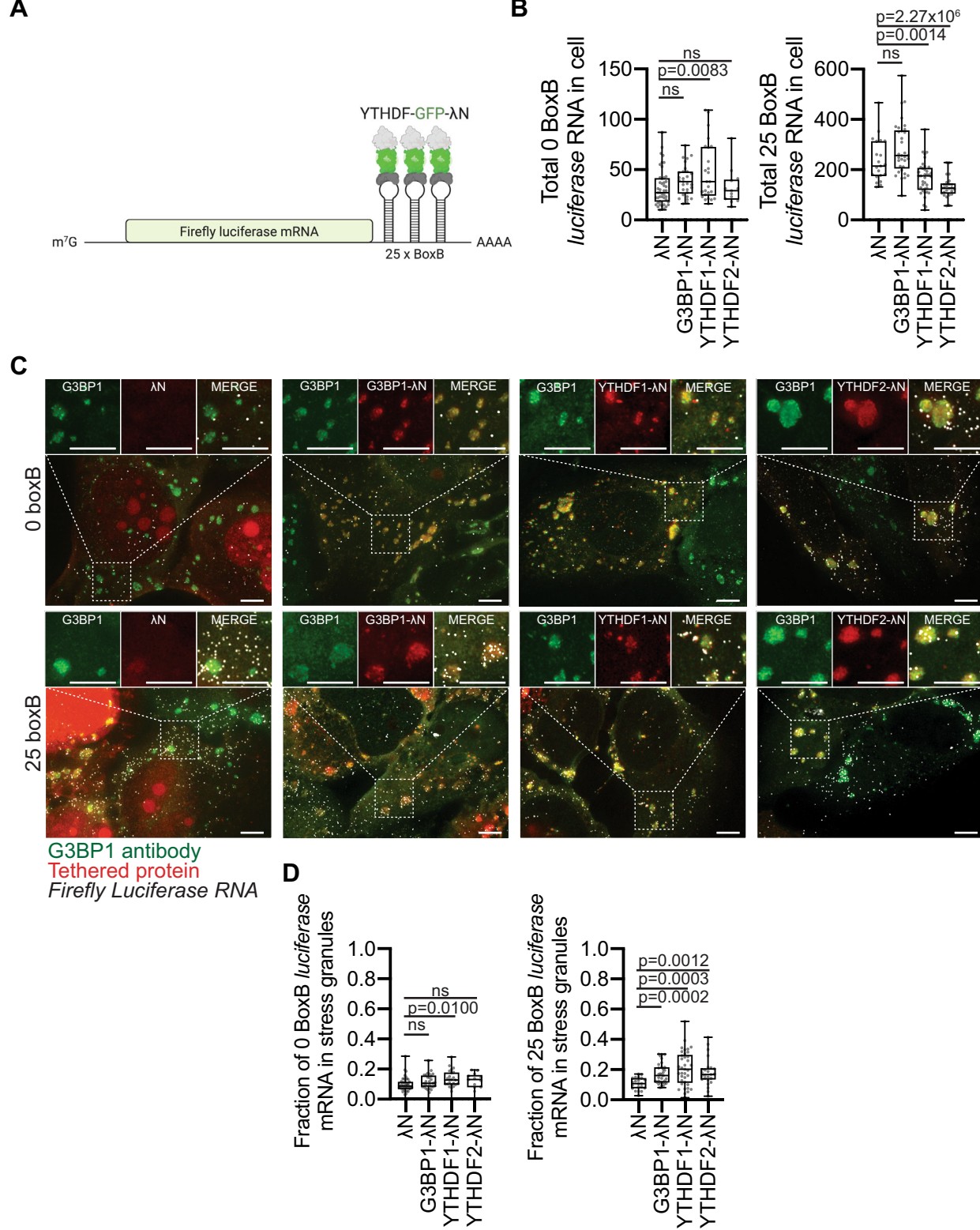

interaction between many mRNAs is reduced by even a small percentage, the assembly curve can shift into the regime where stress granules are not formed.

## Discussion

In summary, we present three observations indicating that m⁶A modification provides at most a minimal role in mRNP targeting

to stress granules. Most importantly, in a direct test of the role of m⁶A modification, we observed that cells deficient in m⁶A modification do not show differences in the partitioning of thirteen m⁶A modified mRNAs into stress granules. We do see a small correlation of m⁶A modification with stress granule enrichment in linear regression analyzes, but in the absence of experimental data showing a role for m⁶A modification in stress granule targeting, it remains possible this correlation is indirect

**Fig. 3 _Luciferase_ RNA is modestly recruited to stress granules when tethered with G3BP1, YTHDF1, or YTHDF2. A** Cartoon schematic of tethering YTHDF proteins to luciferase mRNA using the BoxB-lambdaN (λN) approach. **B** Number of 0 or 25 BoxB _luciferase_ mRNA in U-2 OS cells when expressing λN, G3BP1-GFP-λN, YTHDF1-GFP-λN, and YTHDF2-GFP-λN transgene. U-2 OS cells were stressed for 1 h with arsenite. Each dot represents a cell. $N = 45$, 29, 25, and 13 for λN, G3BP1-GFP-λN, YTHDF1-GFP-λN, and YTHDF2-GFP-λN with 0 BoxB _luciferase_ mRNA. $N = 21$, 33, 37, and 25 for 15 for λN, G3BP1-GFP-λN, YTHDF1-GFP-λN, and YTHDF2-GFP-λN with 25 BoxB _luciferase_ mRNA. The box plots are defined by minima, 25% percentile, median, 75% percentile, and maximum. For λN, the number of 0 boxB RNAs/cell are 10, 18, 27, 41.5, and 87 for minima, 25% percentile, median, 75% percentile, and maximum, respectively. For G3BP1-GFP-λN, the number of 0 boxB RNAs/cell are 16, 26, 38, 48, and 74 for minima, 25% percentile, median, 75% percentile, and maximum, respectively. For YTHDF1-GFP-λN, the number of 0 boxB RNAs/cell are 16, 24, 38, 72.5, and 109 for minima, 25% percentile, median, 75% percentile, and maximum, respectively. For YTHDF2-GFP-λN, the number of 0 boxB RNAs/cell are 13, 20, 29, 40, and 81 for minima, 25% percentile, median, 75% percentile, and maximum, respectively. For λN, the number of 25 boxB RNAs/cell are 131, 174.5, 214, 313.5, and 466 for minima, 25% percentile, median, 75% percentile, and maximum, respectively. For G3BP1-GFP-λN, the number of 25 boxB RNAs/cell are 96, 204.5, 255, 356.5, and 574 for minima, 25% percentile, median, 75% percentile, and maximum, respectively. For YTHDF1-GFP-λN, the number of 25 boxB RNAs/cell are 39, 119.5, 175, 207, and 360 for minima, 25% percentile, median, 75% percentile, and maximum, respectively. For YTHDF2-GFP-λN, the number of 25 boxB RNAs/cell are 56, 104, 126, 145.5, and 228 for minima, 25% percentile, median, 75% percentile, and maximum, respectively. **C** Representative images of arsenite stressed-U-2 OS cells stably expressing 0 or 25 BoxB _luciferase_ mRNA and transiently expressing GFP-λN, G3BP1-GFP-λN, YTHDF1-GFP-λN, and YTHDF2-GFP-λN. Single-molecule FISH probes are illustrated in white, endogenous G3BP antibody staining in green, while tethered proteins are in red. The nuclei are stained with DAPI (blue). The scale bar is 3 μm. **D** Box plot of the fraction of 0 and 25 BoxB _luciferase_ RNA molecules in stress granules in wild-type U-2 OS cells. Each dot represents an experimental data point (or cell). $N = 45$, 29, 25, and 13 for λN, G3BP1-GFP-λN, YTHDF1-GFP-λN, and YTHDF2-GFP-λN with 0 BoxB _luciferase_ mRNA. $N = 21$, 33, 37, and 25 for λN, G3BP1-GFP-λN, YTHDF1-GFP-λN, and YTHDF2-GFP-λN with 25 BoxB _luciferase_ mRNA. The box plots are defined by minima, 25% percentile, median, 75% percentile, and maximum. For λN, the fraction of 0 boxB RNAs in stress granules/cell are 0, 0.068, 0.087, 0.117, and 0.286 for minima, 25% percentile, median, 75% percentile, and maximum, respectively. For G3BP1-GFP-λN, the fraction of 0 boxB RNAs in stress granules/cell are 0, 0.082, 0.105, 0.158, and 0.257 for minima, 25% percentile, median, 75% percentile, and maximum, respectively. For YTHDF1-GFP-λN, the fraction of 0 boxB RNAs in stress granules/cell are 0, 0.095, 0.128, 0.178, and 0.280 for minima, 25% percentile, median, 75% percentile, and maximum, respectively. For YTHDF2-GFP-λN, the fraction of 0 boxB RNAs in stress granules/cell are 0, 0.079, 0.130, 0.160, and 0.194 for minima, 25% percentile, median, 75% percentile, and maximum, respectively. For λN, the fraction of 25 boxB RNAs in stress granules/cell are 0.028, 0.077, 0.108, 0.144, and 0.171 for minima, 25% percentile, median, 75% percentile, and maximum, respectively. For G3BP1-GFP-λN, the fraction of 25 boxB RNAs in stress granules/cell are 0.080, 0.114, 0.160, 0.216, and 0.302 for minima, 25% percentile, median, 75% percentile, and maximum, respectively. For YTHDF1-GFP-λN, the fraction of 25 boxB RNAs in stress granules/cell are 0.015, 0.112, 0.203, 0.298, and 0.519 for minima, 25% percentile, median, 75% percentile, and maximum, respectively. For YTHDF2-GFP-λN, the fraction of 25 boxB RNAs in stress granules/cell are 0.024, 0.134, 0.167, 0.209, and 0.415 for minima, 25% percentile, median, 75% percentile, and maximum, respectively. ns denotes not significant ($P > 0.05$) (unpaired two-tailed student _t_-test). Source data are provided for this Figure (see data availability).

and due to other shared properties. Finally, we observed tethering 25 YTHDF proteins to a reporter mRNA can increase stress granule accumulation. Taken together, these results argue m6A modifications have a small effect on mRNA partitioning, which we estimate as <10%. Modeling the impact of m6A modifications suggests the most optimal situation for m6A to make any difference in stress granule recruitment will be for mRNAs that have limited inherent targeting to stress granules and contain multiple numbers of heavily modified m6A sites (Fig. 2E). For example, one additional m6A site on an RNA with 2500 bases only increases partitioning in stress granules by 1.6%. Thus, our results, in general, argue that m6A is not an epigenetic RNA marker that can direct endogenous mRNA localization to stress granules (Ries et al.).

We note three limitations of our studies. First, there is the formal possibility that the absence of m6A on mRNAs reduces their accumulation in stress granules and that the ΔMETTL3 mES cells have an additional phenotype that increases mRNA accumulation in stress granules by some other mechanism, although the absence of an increase in the partitioning of unmethylated mRNAs into stress granules in the ΔMETTL3 cells makes this unlikely (Supplementary Fig. 2D). Second, the location of the m6A modification on the mRNAs may play a role in stress granule location. For example, Anders et al. proposed stress-induced m6A deposited near the start codon on the 5'UTR direct mRNAs to stress granules during cellular stress with the help of YTHDF3 proteins. However, we do not see changes in stress granule localization for two mRNAs with 5'UTR m6A modifications, _Mtf2 and Pik3r2_, between wildtype and ΔMETTL3 mES cells (Fig. 1, Supplementary Fig. 2B), although we cannot fully rule out stress-induced methylations in the 5' UTR affecting some other mRNAs partitioning into stress granules. Finally, it is

possible that m6A modification might affect mRNA partitioning into stress granules in a different cell type.

Given the limited effect of m6A modification on mRNAs partitioning into stress granules, how can one understand mRNA recruitment into stress granules? Our results with m6A modification are consistent with the model wherein RNA targeting to granules is a summative effect involving many interactions, and no single individual protein-RNA interaction dominates, including m6A-YTHDF interaction[23]. This model explains why RNA length, which is likely highly correlated with valency, which is the number of interactions the mRNA can make with other RNAs and proteins, is so strongly correlated with enrichment in stress granules[9]. This model also explains why length correlation with stress granule enrichment is a consistent metric in other cell types with different genes expressed[9,10]. We suggest that this summative effect may be a general property of RNA targeting to RNP granules because length bias is also seen in the RNAs that accumulate in P-bodies during stress[29], P-granules[30], and BR-Bodies[31]. Therefore, like stress granules, no single individual protein-RNA interactions will substantially affect RNP granule partitioning.

If the m6A modification does not alter RNA partitioning into stress granules, why does the knockdown of multiple YTHDF proteins[26] lead to a decrease in stress granule formation? We suggest the explanation for this observation may be that the IDRs of the YTHDF proteins act as non-specific assembly factors by forming positive interactions with other components of stress granules once the granules are assembled by other specific interactions. This is similar to observations both in vitro and in cells that promiscuously interacting IDRs, which can interact with many other proteins, increase the assembly of condensates by shifting the assembly diagram towards increased assembly once

the promiscuous IDRs are concentrated in the condensate by specific interactions[14]. Thus, we suggest that YTHDF proteins, once bound to mRNAs through m6A modifications, are recruited into stress granules and then through promiscuous interactions with other components of the stress granules, may enhance the assembly of these RNP granules.

## Methods

**Multiple linear regression analysis.** Stress granule transcriptome data was obtained from Khong et al. A reference transcriptome was acquired from GEN-CODE (GRCh38.p13). Reads were preprocessed using trim galore (version 0.6.5). Reads were mapped to the genome using salmon (version 1.8.0). Differential expression analysis was performed using DESeq2 (release 3.15) (Supplementary Data 1). m6A mapped sites, and m6A ratio were obtained from Xiang et al. and Molinie et al. The stress granule enrichment dataset was filtered to only consider the most highly expressed isoform for each gene. The m6A dataset used Refseq annotation, while our original dataset used Ensembl gene ID's. In order to assign m6A peaks to genes in our stress granule isoform data, we matched Refseq annotations to gene names using pybiomart, counted the number of m6A peaks per gene, and merged the two datasets. Transcript lengths were also obtained using pybiomart.

Multiple linear regression analysis was performed using the scikit-learn package in python. Multiple models were constructed using transcript length, the number of m6A sites, the combination as features, and the observed fraction of molecules within stress granules as a response variable. r2 values were obtained to assess the predictive power for each of these models by comparing predicted vs. observed stress granule enrichment. The visualization of our linear model was created by simulating transcripts of varying transcripts lengths up to 7.5 kb in size and of varying numbers of m6A sites and running these data through our multiple linear regression model.

All visualizations in Fig. 2 were created using the matplotlib and seaborn packages in python. All code pertaining to this analysis is contained within the following GitHub repo (https://github.com/tmatheny/m6a/) and linked to Zenodo (10.5281/zenodo.6584757).

**Cell lines.** Wildtype and ΔMETTL3 mES cells were maintained in DMEM supplemented with 15% FCS (ES-009-B, Millipore Sigma), 1X non-essential amino acids (11-140-050, Gibco), 1X penicillin-streptomycin-glutamine (10378016, Thermo Fisher Scientific), 1 mM sodium pyruvate (11360070, Thermo Fisher Scientific), 1X EmbryoMax 2-mercaptoethanol (ES-007-E, Millipore Sigma), 10,000 units/mL ESGRO Leukemia Inhibitory Factor (ESG1106, Sigma-Aldrich), 1 μM PD0325901 (S1036, Selleck Chemicals), and 3 μM CHIR99021 (SML0146, Sigma-Aldrich) and maintained at 37 °C and 5% O2 on 0.1% gelatin (G6144, Sigma-Aldrich) coated tissue culture plates.

U-2 OS cells were maintained in DMEM supplemented with 10% FBS, 1% penicillin/streptomycin at 37 °C, and 5% O2. U-2 OS cells stably expressing 0- and 25-BoxB-luciferase RNA was generated by transducing pLenti-EF1-luciferase-blast and pLenti-EF-luciferase-25-BoxB-blast. The plasmids were created by inserting luciferase and luciferase 25-BoxB derived from plasmids described in Matheny et al. into the pLenti-EF1-Blast vector following Gibson assembly. To generate the Luciferase-25-BoxB and Luciferase lentiviral particles, HEK293T cells (T-25 flask; 80% confluent) were co-transfected with either 2.7 μg of either pLenti–EF1–Luciferase-25-BoxB–blast or pLenti–EF1–Luciferase–blast, 870 ng of pVSV-G, 725 ng of pRSV–Rev, and 1.4 μg of pMDLg–pRRE using 20 μl of Lipofectamine 2000. The medium was replaced 6 h post-transfection. The medium was collected at 24 and 48 h post-transfection and filter-sterilized with a 0.45 μm filter. To generate the U 2-OS luciferase and luciferase-25-BoxB stable cell lines, U 2-OS cells (T-25 flask; 80% confluent) were transduced with 1 ml of lentiviral stocks containing 10 μg/ml of Polybrene for 1 h. The medium was then added to the flask. 24 h post-transduction, the cells were reseeded in T-25 flask containing 5 μg/ml of blasticidin selective medium. Single colonies were selected, and colonies with an adequate expression of RNA were chosen for this project (20–500 mRNA copies per cell).

**Absolute quantification of m6A levels on mRNAs in wildtype and ΔMETTL3 mES cells.** Total RNA was isolated from wildtype and ΔMETTL3 mES cells using Trizol reagent (15-596-018, Thermo Fisher Scientific) following the manufacturer's protocol. Poly(A) mRNA was then extracted using Dynabeads mRNA purification kit (61006, Thermo Fisher Scientific) following the manufacturer's protocol. Absolute quantification of m6A levels on poly(A) mRNA was determined using Epiquick m6A RNA methylation quantification kit (P-9005-48, EpiGentek) following the manufacturer's protocol. Biological replicates were performed to confirm results are reproducible.

**m6A-immunoprecipitation and qRT-PCR.** The m6A-immunoprecipitation protocol was adapted[32].

Total RNA was extracted from wildtype and ΔMETTL3 mES cells in 10 cm plates using Trizol extraction. The RNA was then fragmented using NEBNext

Magnesium RNA fragmentation module (E6150S; NEB) following the manufacturer's protocol. The reaction was cleaned using RNeasy Mini kit (74106; Qiagen) following the manufacturer's protocol.

m6A RNA was then precipitated in the following manner. First, 25 μL of protein G (#S1430, NEB) and protein A dynabeads (10001D, Thermo Fisher Scientific) are prepared by washing beads twice with 100 μL 1x PP buffer (150 mM NaCl, 0.1% NP-40, and 10 mM TrisHCl (pH 7.5)) and tumble overnight in 1x IPP buffer with 3 μg of rabbit m6A antibody (202 003; Synaptic Systems).

6 μg of fragmented RNA was heat-denatured at 70 °C for 2 mins. The RNA volume was then matched with 2x IPP. 1x IPP buffer was then added to get a total volume of 150 μL.

The protein G and protein A dynabeads-antibody mix were washed twice with 100 μl 1x IPP buffer and resuspended in 50 μl 1x IPP buffer with 1 μL Ribolock (EO0382; Thermo Fisher Scientific). Subsequently, the RNA samples are added to the protein G-antibody mix and tumble rotated at 4 °C for 3 h.

The dynabeads-antibody-RNA mix is then washed with several different buffers. First, the mix is washed twice with 1x IPP buffer. Then it is washed twice with low salt buffer (50 mM NaCl, 0.1% NP-40, and 10 mM TrisHCl (pH 7.5)). And washed twice again with high salt buffer (500 mM NaCl, 0.1% N-40, and 10 mM TrisHCl (ph 7.5)). Finally, the mix was washed once with 1x IPP.

After washes, the mix was eluted by adding 50 μL RLT buffer (74106; Qiagen). The eluate is transferred to a new tube where three volumes of 100% ethanol are added and mixed. The mixture is added to the RNeasy column, spun, and washed once with 1x buffer RPE and subsequently spun dried once. The RNAs were then eluted twice with 50 μL water.

2x IPP solution was then added to the eluate to bring the volume to 200 μL. The IP/washes/RNeasy cleanup was performed a second time with the protein A-antibody beads and eluted twice with 50 μL water.

500 ng of total RNA and 11 μL of IP-eluted RNA samples were used for qRT-PCR reaction. cDNA was made using Invitrogen Superscript III Reverse Transcriptase (18-080-093, Thermo Fisher Scientific) by following the manufacturer's protocol. qRT-PCR was performed using iQ SYBR Green Supermix following the manufacturer's protocol (170880, Bio-Rad). The primers used for qRT-PCR are listed in Supplementary Data 2.

**Generation and transfection of GFP-λN, G3BP1-GFP-λN, YTHDF1-GFP-λN, and YTHDF2-GFP-λN plasmids.** The creation of GFP-λN and G3BP1-GFP-λN plasmids were previously described in Matheny et al. YTHDF1 and YTHDF2 were synthesized by Twist Biosciences and subcloned into G3BP1-GFP-λN vector by swapping out G3BP1 with YTHDF1 and YTHDF2 genes using EcoR1 and BamH1 sites. Inserts were sequenced and verified.

**Sequential immunofluorescence and single-molecule FISH.** U-2 OS, wildtype and ΔMETTL3 mES cells were seeded on EtOH-sterilized, +0.1% gelatin-coated (mES cells), 18 × 18 mm #1.5 coverslips in six-well tissue culture plates overnight. Cells were stressed by replacing the media with media containing 500 μM NaAsO2 and incubating for 1 h at 37 °C/5% CO2, followed by washing with prewarmed 1X PBS and fixing with 500 μl of 4% paraformaldehyde for 10 min at room temperature. After fixation, cells were washed twice with RNase-free 1X PBS (10010049, Thermo Fisher Scientific), permeabilized with 0.1% Triton X-100 in RNase-free 1X PBS for 5 min, and washed once with RNase-free 1X PBS.

Cells were either stained with antibodies first (Supplementary Fig. 2C–D, Fig. 3) or smFISH probes first (Fig. 1, Supplementary Fig. 1A–B). A fixation step with 500 μL of 4% paraformaldehyde for 10 min was performed between the antibody and smFISH staining (and vice versa). For antibody staining: cells were stained with primary stress granule antibodies (5 μg/mL mouse α-G3BP primary antibody (ab56574; Abcam) or 1 μg/mL Rabbit α-PABP primary antibody (ab21060; Abcam)) in RNase-free 1X PBS for 60 mins at room temperature, followed by three washes with RNase-free 1X PBS, and subsequently stained with respective secondary antibodies (1:1000 goat α-mouse FITC-conjugated secondary antibody (ab6785; Abcam) or 1:1000 donkey anti-rabbit Alexa-fluor 555 antibody (ab150062, Abcam)) in RNase-free 1X PBS for 60 mins at room temperature. Coverslips were then washed three times with RNase-free 1X PBS and fixed with 500 μl of 4% paraformaldehyde for 10 min at room temperature. For single-molecule FISH staining: the protocol was adapted from[33]. Cells were incubated in Wash buffer A (10% deionized formamide) for 5 mins at room temperature. Cells were then incubated in hybridization buffer (10% dextran sulfate, 10% deionized formamide in 2X SSC) with 125 nM smFISH probes for 15 h at 37 °C in a humidifying chamber. Cells were then incubated twice with Wash Buffer A for 30 mins each at 37 °C and once with Wash Buffer B (2X SSC) for 5 mins at room temperature. The coverslips were then mounted on slides with Vectashield antifade mounting medium with DAPI (101098-044, VWR).

Single-molecule FISH probes, except for the POLR2A (SMF-2006-1, Biosearch Technologies) and firefly luciferase mRNA single-molecule FISH probes[23] are now created using a method as described in Gaspar et al.[34]. The oligo sequences can be found in Supplementary Data 2. Thirty 200 μM oligos were mixed together in one Eppendorf (master mix). And oligos were then labeled with ddUTP-Atto633 (JBS-NU-1619-633, Axxora) or -Atto555 (NU-1619-500, Jena Bioscience) by combining 5 μL of master mix, 1 μL of ddUTP-Atto633 or -Atto555, 0.3 μL of TdT enzyme, 3 μL of TdT buffer (EP0161, Thermo Fisher Scientific), and 5.7 μL of H2O in a PCR

tube. The reaction was incubated at 37 °C for 16 h in a PCR machine and fresh. In all, 3 μL or TdT enzyme was added 8 h into reaction. Oligo sequences can be found in Supplementary Data 2. The reaction was purified using an oligo clean & concentrator kit following the manufacturer's protocol. Labeled oligonucleotides were then analyzed by absorption at 629 nm (which is the absorption max for Atto 633) or 554 nm (which is the absorption max for Atto554). The concentration of labeled oligos was then determined following Beer's law and diluted to a stock concentration of 12.5 μM.

Imaging parameters are adapted from previously in Khong et al. for Supplementary Fig. 2C–D and Fig. 3. Samples were imaged on a GE wide-field DeltaVision Elite Microscope equipped with an Olympus UPlan-SApo 100X/1.40-NA Oil Objective lens and a PCO Edge sCMOS Camera using appropriate filters with the help of SoftWoRx Imaging Software. Entire cells were imaged using an approximate number of Z sections (0.2 μm step size). Imaging parameters were adjusted to capture fluorescence within the microscope's dynamic range and kept the same between samples when looking at the same mRNAs by smFISH. Standard deviations were determined from three biological replicates in Supplementary Fig. 1E–F. Each data point in Fig. 3 represents one cell. Only cells that were adequately expressing the transgene GFP-λN, G3BP-GFP-λN, YTHDF1-GFP-λN, and YTHDF2-GFP-λN were counted (50,000 < Cell Total Cell Fluorescence <2,000,000 (determined by ImageJ)) in Fig. 3. The images were blinded, and the single-molecule FISH spots were manually counted. Total numbers of spots outside and inside stress granules were determined. All images shown in the manuscript are deconvolved, and the brightness/contrast adjusted to best indicate data.

Images for Fig. 1 and Supplementary Fig. 1C–D were taken on a laser scanning confocal microscopy (Nikon A1R microscope) with ×100 objective (1.45 NA, Plan Apo I), a pixel size of 0.10 μm, and an integration time of 2.2 μsec. Other imaging parameters were adjusted to capture fluorescence within the microscope's dynamic range and kept the same between samples when looking at the same mRNAs by smFISH. Image analysis for Fig. 1 and Supplementary Fig. 1C–D was performed using Imaris (Version 9.7.0). Only smFISH spots in the cytoplasm were counted. smFISH in the nucleus was excluded from the analysis. We used Imaris to classify smFISH spots, stress granules and nuclei, and smFISH spots in stress granules and nuclei. Image analysis parameters for the same smFISH was done equally between wildtype and ΔMETTL3 mES cells. Since mES cells have a small cytoplasm where many of the stress granules are touching the nuclei, we notice smFISH can be artificially double counted in both stress granules and nuclei by Imaris. When this occurs, we call these smFISH spots in nuclei and not in stress granules. We also noticed a small number of artificial smFISH spots that were observed outside mES cells/colonies. We subtracted these false-positive spots from our analysis. Standard deviations were determined from three biological replicates in Fig. 1 and Supplementary Fig. 1E–F.

**Reporting summary**. Further information on research design is available in the Nature Research Reporting Summary linked to this article.

## Data availability
Quantitative source data for Figs. 1, 2A, E, F, 3B, D, Supplementary Fig. 1A–D, and Supplementary Fig. 2A–D in this study are provided as a Source data file. Quantitative source data for Fig. 2B–D and Supplementary Fig. 3 are deposited at https://github.com/tmatheny/m6a/. The github repository is linked to Zenodo (https://doi.org/10.5281/zenodo.6584757). Raw imaging files that were included in the figures are deposited at Figshare.com (https://doi.org/10.6084/m9.figshare.19780576.v1). Not all raw imaging files (used for analysis) were included due to the number and sizes of images. These raw imaging files can be shared upon request. Source data are provided with this paper.

## Code availability
All code pertaining to this analysis is contained within the following GitHub repo (https://github.com/tmatheny/m6a/) (https://doi.org/10.5281/zenodo.6584757).

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

## Acknowledgements

We like to thank Pedro Batista (NIH) for providing us with both wildtype and knockout *METTL3* mES cells and for providing instructions on how to maintain the cells and how to do m6A-immunoprecipitation. We like to thank Nancy Kedersha (Brigham and Women's Hospital) for providing us with U-2 OS cells. We are grateful to Joe Dragavon (BioFrontiers Institute Advanced Light Microscopy Core) for training on the laser scanning confocal microscopy (Nikon A1R microscope), and image analysis training. We like to thank John Rinn (CU Boulder) for providing FCS for growing mES cells. Finally, we like to thank Ye Fu and Xiaowei Zhuang (Harvard University) for stimulating and helpful discussions. This work was supported by Banting Postdoctoral Fellowship (A.K.) and Howard Hughes Medical Institute (R.P.).

## Author contributions

A.K., T.M., and R.P. conceptualized the study. A.K., T.M., and T.H. developed and designed the methodology. A.K. and V.B. performed experiments and collected data. A.K. and T.M. performed formal analysis. A.K. and R.P. wrote and edited the manuscript. A.K. and R.P. supervised the project. A.K. and R.P. provided funding.

## Competing interests

R.P. is a founder and consultant for Faze Medicines. Other authors do not have competing interests.
