## [Peer Review File · Nature Communications]

Title: Limited effects of m6A modification on mRNA partitioning into stress granulesREVIEWER COMMENTS

Reviewer #1 (Remarks to the Author):

This manuscript presented data to show that m6A has a smaller effect than previously reported to localize mRNA to SG. I read this manuscript as well as previous reports on this subject, I can clearly see previous overstatement. I do have a few comments to be more balanced.

Page 3, Line 58-61, Page 5, Line 104-107, and Fig 1B, C. There is still 25% of m6A in METTL3 KO mES cells. It is unclear how much m6A are still there in the specific mRNA transcripts analyzed in the METTL3 KO mES cells. In addition, this experiment alone is not enough to conclude that m6A has no effect in recruiting mRNA to SGs in other cell lines. m6A depletion could increase the stability and steady-state level of many mRNA, and can also change the alternative polyA sites. The consequences of these effects should be considered or discussed.

Effects of m6A is very cell dependent. I would strongly urge the authors to repeat experiments in Fig. 3B and 3C in HeLa cells. I would like to see results comparing to current cell line.

Page 7, Line 131-137: The author described the increase from 0.38 to 0.40 for m6A (Fig 2D), which has not considered the effect of m6A ratio. A more precise description should be "the increase from 0.41 to 0.47" (Fig S1A and S1F). And these figures could be switched with Fig 2D and moved to the main figure panels. An increase from 0.41 to 0.47 might not be minimum.

Page 7 line 144-146, and Fig 2E: The simulation uses ORF length between 1 to 30 kb. However, in cellular mRNA, CDS length is mostly smaller than 8000 (only 1% of mRNA have CDS larger than 8000, 3.5% larger than 5000). As predicted by the chart, with 0 mRNA site, from 2k to 8k CDS length, SG ratio increases from around 10% to 40% in SG, which is a 30% increase.

Fig. 2E. To better demonstrate the m6A effect, the Y-axis could be a m6A factor set to be the linear combination of m6A sites (from 0 to 8) and m6A ratio (from 0 to 100%) with parameters determined by the multi-linear regression model, and X-axis could be mRNA length (300 to 15 kb) or ORF length (200 to 10 kb).

Page 8 line 154-155, in the linear regression analysis, the m6A site is not assumed to be 100% modified. The linear regression model has already counted in the stoichiometry of m6A in the analysis. Line 34, "and those sites are not fully modified" is not true as well. As such, the claim that the simulation is likely overestimating the contributions of m6A (Line 151) is also not true. Page 8, line 159-161. Figure 2G: "Even assuming 100% modification of each site". Again, this is not true. Since 6.8% is estimated using the average m6A stoichiometry of the individual site, a new number could be calculated from the multi-linear regression model by taking into account both m6A site and m6A ratio.

Page 9, Line 189-192 and Fig 3. the increase from 11% to 21% is not "a small effect" considering it is

almost a 2-fold change. In addition, the fraction of endogenous mRNA transcripts within stress granules are mostly between 0% to 20%, with an average of around 11%. Almost no mRNA can go above 40% (as can be observed from the histogram on Fig 2B). These data suggest that the dynamic range of fraction of mRNA within SG is limited to mostly between 0% and 40%, as such, an increase from 11% to 21% is still quite significant.

Fig 3D: It is unclear how many repeats of experiments were performed.

Overall, the data presented suggests that the effect of m6A likely accounts for 15% to 20% as compared to the effect of mRNA length. The effect is there but not large in cell lines the authors studied. I am curious about HeLa cells.

Reviewer #2 (Remarks to the Author):

In the manuscript “Limited effects of m6A modification on mRNA partitioning into stress granules”, Khong et al. analyze the effect of m6A modification on the recruitment of mRNAs into stress granules. Two studies (Anders et al, 2018; Ries et al, 2019) showed that m6A RNA modification has a role in the recruitment of RNAs to SGs through interaction with YTHDF proteins. Specifically, Anders et al showed the recruitment to be YTHDF3 dependent through stress-induced m6A in the 5' vicinity of the CDS. Here, the authors use smRNA FISH targeting Fem1b and Figl1 mRNAs on mES WT cells and methylation writer Mettl3 KO cells and find no differences in their recruitment to SGs. Linear regression analysis using annotated m6A sites (Xiang et al 2017) and RNAs recruited to SGs (Khong et al 2017) shows that overall m6A contributes slightly to SG recruitment. YTHDF1 and 2 tethering assays reveal an increase in the recruitment to SGs. Overall, using a combination of these approaches, the authors find that m6A modification and YTHDF binding play a role in stress granule recruitment but a minor one, which contradicts the findings of the two previous studies.

The topic of what RNA features affect their recruitment to stress granules has been a subject of extensive studies and it is of great interest to the community. The experimental results presented here are thought provoking. However, it is important to note that although the authors refer to two previous studies (Anders et al, 2018; Ries et al, 2019), the model proposed by Anders et al is not addressed: i) the location of stress-induced m6A is not considered and ii) YTHDF3 contribution to SG recruitment is not addressed. My overall recommendation is to further develop the study to strengthen the conclusions. A more robust analysis of the relationship between m6A and SG recruitment is needed. However, I am afraid that unless the conclusions are different from previous studies, the work might in general lead to repetition of previous works.

Major comments:

1. The two previous studies show that m6A modification enhances or facilitates the recruitment of mRNA to stress granules, while Khong et al show that m6A modification has a limited effect.

Quantitatively speaking, it is difficult to discern enhancing, facilitating or limiting. I would suggest that the comparison is done more quantitatively and clearly.

2. In general, both studies (Anders et al, 2018; Ries et al, 2019) are referred to as showing that overall m6A aids in the recruitment to SGs through interaction with YTHDF proteins. Nevertheless, these two studies are conceptually different and propose different models of m6A contribution to SG recruitment, and the authors address only the Ries et al model.

3. The original study by Batista et al found that Mettl3 KO results in ~60% reduction of m6A in Mettl3 KO mESC. smRNA FISH convincingly shows that both RNAs are recruited to SGs in Mettl3 KO, but supporting evidence is needed. Is the methylation of Fem1b and Figl1 RNAs reduced? Is there a gain of methylation upon stress? Please complement the smRNA FISH experiments by analysis of Fem1b and Figl1 methylation using RIP or a similar approach.

4. Several questions remain regarding the two genes chosen for smRNA FISH. It is not clear why Fem1b and Figl1 mRNAs were chosen, apart from their use by Ries et al, and the choice does not seem drawn in a representative manner. Are Fem1b and Figl1 methylated in mES cells where RNA FISH was performed? Since the conclusions are extrapolated to a global scale, I would suggest increasing the pool of RNAs to those that are a representative subset (methylated, interact with YTHDF proteins, recruited to SGs and analyze their recruitment in the absence of methylation). In making this choice, the authors should bear in mind the location of m6A sites, induction of m6A by stress and YTHDF1, 2 or 3 binding.

5. It was shown that RNAs with stress-induced m6A around the 5' vicinity of their CDS (not overall methylation) are recruited to SGs by interacting with YTHDF3. Where do the m6A sites of these two mRNAs reside? The linear regression model and smRNA FISH assay do not consider the relative location of the m6A within the transcript. An additional caveat with the linear regression analysis is that it uses datasets from different studies, thus the methylation status of RNA might be different. The authors should consider the pattern of m6A methylation in their analyses to be consistent with the model they are addressing.

6. Are YTHDF proteins recruited to SGs in mES WT and Mettl3 KO cells?

7. YTHDF tethering assay is performed to determine whether YTHDF proteins have a role in the recruitment of mRNA to SGs. I have two concerns about the assay. First, there are a total of 25 binding sites. As authors note, this is beyond the average number of m6A sites in endogenous RNAs, which leads them to conclude that the observed positive effect is due to a larger number of binding sites and likely much lower for endogenous RNAs. This raises the question of why 25 binding sites in the first place? Why not a number closer to biological levels or to determine the linearity of the effect? In order to place the conclusions in a biological context, the number of binding sites in the tethering assay should be closer to what authors are trying to address. Secondly, the authors use YTHDF 1 and 2 binding assay, while Anders et al propose a model wherein mRNAs associate with SGs by stress induced methylation in a YTHDF3-dependent manner. Considering that they are directly addressing that study, I would suggest the expansion of tethering experiments to include YTHDF3.

8. The direct evidence on a global scale that would show that m6A does not play a role in SG recruitment is to analyze the RNA content and RNA methylome of SGs in WT and KO cells.

Minor comments:

1. Lines 26, 60 and similar: please mention the two mRNAs for which this was shown by smRNA FISH;

the current formatting can be easily understood as all m6A-modified RNAs.

2. Fig 1B and Fig 3 – Please note what each data point represents in the figure. It is important to note how many individual cells and stress granules were quantified because the total number of FISH spots is not informative enough (we do not know how many FISH spots one cell has). One biological replicate should have a representative set of cells/SGs quantified.

3. Fig 3C – how many cells and biological replicates?

4. Tethering assay. When showing the relative portion of RNA in SGs per cell, the representative images would be better interpretable if the image would highlight the boundaries of one cell. Is the amount of luciferase RNA constant between different samples?

5. As m6A modification is implicated in regulating RNA metabolism, was the total number/stability of Fem1b and Figl1 mRNAs affected by the loss of methylation?

6. The results in Fig. 3D, as authors note in the manuscript, show that YTHDF1 and 2 tethering has an effect on recruitment to SGs. Thus, it is puzzling that this effect is disregarded for the number of used binding sites, without knowing the linearity of the effect.

7. Line 181 – please note this is the average \pm

8. Line 189, 191 – “small effect”, “small role”, please use a quantitative description of the data, while the interpretation if this is a small or physiologically relevant contribution in the discussion.

9. Line 344 – Figure 3 not 2

Reviewer #3 (Remarks to the Author):

Stress granules are cytoplasmic membrane-less condensates that are formed in diverse stress conditions. The condensate is comprised of non-translating mRNPs. Although the protein components of stress granules were around, the mRNA content began to be revealed only recently (Khong et al., 2017; Namkoong et al., 2018). A major conclusion from these studies is that longer mRNAs tend to localize more in the stress granule. Despite the length difference, mRNAs of the similar length show different recruitment into stress granules, raising a possibility of sequence-dependent targeting of mRNA transcripts. Indeed, recent works (Anders et al., 2018; Ries et al., 2019) showed that m6A modifications promote mRNA recruitment to stress granules through the interaction with YTHDF proteins. In this paper, Khong et al. argues, based on three major reasonings listed below, that m6A modifications are unlikely to play a significant role in recruiting mRNAs to stress granules.

First, the authors examined the localization of two poly-m6A mRNAs, FIGNL1 and FEM1B, in wildtype

and Δ METTL3 mES cells in stress granules and found that these two mRNAs are enriched similarly in stress granules in both cell types. These particular mRNAs are examples of polymethylated mRNA and of showing higher smFISH localization signals in the stress granule compared to non-methylated mRNA (Ries et al., 2019). Since METTL3 is a component of m6A methylase, based on their observation, the authors claimed that m6A does not play a major role in the localization of mRNAs into stress granules. Second, the authors performed multiple regression analyses where they compared the effect of mRNA length on stress granule targeting with or without an additional contribution of m6A modifications. Their analyses showed that the ORF length is a much stronger predictor for mRNA partitioning than m6A modifications, and for fixed transcript lengths, m6A modifications up to 4 m6A lead to an increase in predicted stress granule enrichment from $\sim 15\%$ (0 m6A) to $\sim 22\%$ (4 m6A).

Third, they used an artificial tethering assay based on the λ N-BoxB system to ask if the tethering of YTHDF proteins on the reporter mRNA leads to recruitment of the mRNA into the stress granule. Based on the change in the recruitment from 11% to $\sim 20\%$, they claimed that most endogenous mRNAs with smaller number of m6A sites than the reporter mRNA are unlikely to be affected by m6A modifications in the recruitment to stress granules.

I think these reasonings are poorly supported as I discuss in detail below. I suggest that the authors focus on providing extra transcriptome-level localization data in Δ METTL3 mES cells since other two datasets (regression analysis and artificial tethering assay) are only indirect evidence and subject to different interpretations.

Major concerns

1. The smFISH experiment in Δ METTL3 mES cells: In this experiment, their conclusion is drawn from the localization pattern of just two mRNAs, FIGNL1 and FEM1B, which shows a minimal change upon the knock-out of METTL3. In Fig 1A, they reported that the amount of m6A is reduced overall by $\sim 70\%$ in Δ METTL3 mES cells. However, the degree of reduction in m6A modification in the mRNAs being tested is unknown. Without a detailed analysis on m6A levels for these transcripts, it is hard to make a conclusion regarding the effects of m6A modifications on the localization of particular transcripts. In addition, the number of mRNAs tested, 2 here, is highly limited. In fact, previous works (Fig 4B in Ries et al., 2019 and Fig 1B in Fu & Zhuang, 2020) already showed that the m6A level in a given mRNA is not a determining factor for stress granule recruitment but provides only an overall tendency for the recruitment: For a given number of m6A, stress granule enrichments are broadly distributed with significant overlaps between neighboring populations. I thus suggest that the authors examine the localization change in a much larger group of mRNAs. The transcriptome-level analysis would provide unambiguous evidence to test their claim.

2. Multiple regression analyses: In Fig 2E and 2G, the regression model predicted that the fraction of 2.5kb RNA within stress granules increases from 15% (0 m6A) to 22% (4 m6A) as the m6A modification varies. However, the way data is present is misleading. When a given mRNA without any m6A is modified to have 4 m6A sites, the predicted change of the mRNA fraction in the stress granule is in fact $\sim 50\%$ increase ($= (22-15)/15$) from the original value. It should be noted that the overall fraction of bulk mRNA within the stress granule is very low ($\sim 10\%$) (Khong et al., 2017; Namkoong et al., 2018). Moreover, the length of mRNA is not a major target of regulation once the transcript is formed. From the perspective of localization regulation, the predicted change may not be a small amount. I suggest

that rather than the model prediction of which result may be differently interpretable, the authors focus on providing extra experimental evidence to support their claims.

3. BoxB- λ N reporter assay: The data seem solid but regarding interpretation I still find the same issues as above. The authors indeed observe an increase in the stress granule partitioning of the reporter mRNA from 11% to 21% when tethered by YTHDF. This corresponds to almost 100% increase compared to the untethered case.

Minor points

1. (In Fig 1C legend) Are these data from U2OS cells? If they are not typos, I suggest they repeat this analysis for mES cells for consistency. Also, the number of images/cells analyzed must be specified.
2. (Fig 1B) PABP -> G3BP?
3. (page 5, line 89-91) This sentence needs to be modified.

OVERVIEW OF RESPONSES

We thank the reviewers for their comments. Their suggestions have led us to fundamentally improve the manuscript. Most importantly, we now examine the partitioning of 13 different mRNAs to stress granules in WT and METTL3 Δ mES cell lines by smFISH, 11/13 that have been demonstrated to show reduced methylation in METTL3 Δ cells (Batista et al., (2014)). The critical observation is that none of these mRNAs shows decreased accumulation in stress granules when m⁶A methylation is reduced. This direct test of the model that m⁶A modification targets mRNAs to stress granules, the increased number of mRNAs examined, and the consistent absence of any effect, allows us to be confident in our conclusions.

We have also improved the presentation of our correlation analysis, and the artificial situation where YTDHF proteins are tethered to mRNAs. The results of these experiments reinforce our conclusions, but since these are artificial, or strictly correlative, analyses they are not as central to the manuscript as the smFISH data in WT and METTL3 Δ cell lines.

Taken together, we significantly strengthened this manuscript and it is now ready for publication in Nature Communications.

SPECIFIC RESPONSES TO REVIEWER COMMENTS

Reviewer #1 (Remarks to the Author):

This manuscript presented data to show that m⁶A has a smaller effect than previously reported to localize mRNA to SG. I read this manuscript as well as previous reports on this subject, I can clearly see previous overstatement. I do have a few comments to be more balanced.

Page 3, Line 58-61, Page 5, Line 104-107, and Fig 1B, C. There is still 25% of m⁶A in METTL3 KO mES cells. It is unclear how much m⁶A are still there in the specific mRNA transcripts analyzed in the METTL3 KO mES cells. In addition, this experiment alone is not enough to conclude that m⁶A has no effect in recruiting mRNA to SGs in other cell lines. m⁶A depletion could increase the stability and steady-state level of many mRNA, and can also change the alternative polyA sites. The consequences of these effects should be considered or discussed.

In this comment, the reviewer raises two issues that we address in turn.

First, the reviewer points out that we do not know how much the methylation of the Fem1b and Fgn11 mRNAs is reduced in the METTL3 KO mES cells. This is a valid point and to address this issue we have examined 11 additional mRNAs known to have reduced methylation in METTL3 KO mES cells compared to wildtype mES cells (Batista et al., (2014)). The stress granule localization of all 11 of these m⁶A mRNAs are not statistically different between wildtype and METTL3 KO mES cells. In fact, unexpectedly, 9 out of 11 mRNAs show slightly increased stress granule localization in METTL3 KO mES cells. This reinforces our conclusion that m⁶A methylation does not promote the localization of mRNAs into stress granules.

Second, the reviewer makes the point that there could be alterations in RNA stability and/or RNA processing that would “hide” the effect of m6A playing a role in recruiting RNA to stress granules. However, changes to mRNA stability should not affect our analysis since we standardize every assay to the fraction of each mRNA accumulating in stress granules. To acknowledge this possibility, we do add a discussion of this limitation in the manuscript. However, since the original arguments for m6A localizing mRNAs to stress granules were flawed (see manuscript introduction), and the direct test of the model shows no effect of m6A modification on stress granule recruitment on all mRNAs examined, the parsimonious explanation is that m6A modification does not affect the partitioning of mRNAs into stress granules

Effects of m6A is very cell dependent. I would strongly urge the authors to repeat experiments in Fig. 3B and 3C in HeLa cells. I would like to see results comparing to current cell line.

Our paper is focused primarily on mES cells for two reasons. First, to our knowledge they are the only cell line that is viable without METTL3. Indeed, we obtained U-2 OS cells previously described as lacking in methylation (Xiang et al., 2017, Nature), but our experiments revealed these cells do not show alterations in m6A methylation (data not shown), thus limiting our ability to examine other cell types. Second, we used mES cells since these cells were used by Jaffrey’s laboratory in Ries et al., (2019). It is possible other cell lines may show m⁶A dependence for RNA recruitment to stress granules, but we think this is highly unlikely based on our understanding of stress granule assembly, and our observation that m6A methylation does not matter when experimentally examined. Given the toxicity of METTL3 depletion in HeLa cells we have not added this experiment but do acknowledge the possibility of cell type specific effects in the discussion.

Page 7, Line 131-137: The author described the increase from 0.38 to 0.40 for m6A (Fig 2D), which has not considered the effect of m6A ratio. A more precise description should be "the increase from 0.41 to 0.47" (Fig S1A and S1F). And these figures could be switched with Fig 2D and moved to the main figure panels. An increase from 0.41 to 0.47 might not be minimum.

We have corrected this error as suggested by the reviewer. However, we do note that these types of analyses are strictly correlative and need to be supported by laboratory experimentations. For example, Ries et al., (2019) Nature shows strong correlation between # of m⁶A mapped sites and stress granule enrichment. However, we now know that this is largely an artifact because both parameters correlate strongly with length. Therefore, we concentrated our effort on looking at the localization of more m6A RNAs by smFISH in this revision. We saw no significant change in stress granule partitioning of mRNAs between wildtype and METTL3 KO mES cells (Figure 1), which supports our weak correlation data.

Page 7 line 144-146, and Fig 2E: The simulation uses ORF length between 1 to 30 kb. However, in cellular mRNA, CDS length is mostly smaller than 8000 (only 1% of mRNA have CDS larger than 8000, 3.5% larger than 5000). As predicted by the chart, with 0 mRNA site, from 2k to 8k CDS length, SG ratio increases from around 10% to 40% in SG, which is a 30% increase.

We agree with the reviewer that it would be more appropriate to limit Figure 2E to mRNAs less than 10 kb, which we have now done.

Fig. 2E. To better demonstrate the m6A effect, the Y-axis could be a m6A factor set to be the linear combination of m6A sites (from 0 to 8) and m6A ratio (from 0 to 100%) with parameters determined by the multi-linear regression model, and X-axis could be mRNA length (300 to 15 kb) or ORF length (200 to 10 kb).

In this comment, the reviewer is suggesting alternative ways we could develop a model for our simulations. While we agree, different approaches could be used here, we believe the approach we have used is valid. More importantly, the key data in this manuscript is the experiments demonstrating there is no difference in mRNA partitioning into stress granules when m6A modifications are reduced. The modeling is secondary to this critical experimental evidence.

Page 8 line 154-155, in the linear regression analysis, the m6A site is not assumed to be 100% modified. The linear regression model has already counted in the stoichiometry of m6A in the analysis. Line 34, "and those sites are not fully modified" is not true as well. As such, the claim that the simulation is likely overestimating the contributions of m6A (Line 151) is also not true. Page 8, line 159-161. Figure 2G: "Even assuming 100% modification of each site". Again, this is not true. Since 6.8% is estimated using the average m6A stoichiometry of the individual site, a new number could be calculated from the multi-linear regression model by taking into account both m6A site and m6A ratio.

In this comment, the reviewer makes two points that we clarify in turn. First, he/she points out that in our initial linear regression model, the stoichiometry of the m6A modification is already included in the analysis since m6A ratio is part of the model. Therefore, in this analysis each m6A site is not assumed to be 100% modified. This is accurate and we have rephrased the text accordingly.

In this second comment, the reviewer extends this analysis to the modeling in Figure 2E. The model for these figures is solely based on the number of m6A sites, and to reveal the maximum possible effect of m6A modification, we assumed each site is 100% modified. We also built a new model that now incorporates m6A ratio as a third variable (Reviewer Figure 1).

Reviewer Figure 1: Three-way model of predicated fraction of transcripts within stress granules with the following metrics: overall transcript length, m6A mapped sites and m6A ratio. An interactive file is included (file name: 3d_scatter.html).

We don't think this is worth including since a) assuming the modification occurs at 100% shows the maximum predicted effect, and since it is small it provides correlative data that m6A modification is not a major determinant of targeting mRNAs to stress granules, which is strongly demonstrated by our experimental data (Figure 1 and supplemental Figure 1).

Page 9, Line 189-192 and Fig 3. the increase from 11% to 21% is not "a small effect" considering it is almost a 2-fold change. In addition, the fraction of endogenous mRNA transcripts within stress granules are mostly between 0% to 20%, with an average of around 11%. Almost no mRNA can go above 40% (as can be observed from the histogram on Fig 2B). These data suggest that the dynamic range of fraction of mRNA within SG is limited to mostly between 0% and 40%, as such, an increase from 11% to 21% is still quite significant.

We note that in order to get a 2X increase of mRNA in stress granules, 25-tethered YTHDF proteins were needed. However, most human mRNAs have less than 4 m6A sites arguing that the effect of m6A on the vast majority of mRNAs will be much lower or non-existent, which is what we observe in the direct experimental test of the role of m6A on stress granule targeting. Moreover, the tethering assay may have significant drawbacks that would likely overemphasize the role of recruiting m6A RNA to stress granules. (1) First, the Kd for BoxB elements binding to LambdaN proteins are in the order of ~1.3nM (Keryer-Bibens et al., (2008)). This is approximately 500-fold greater than the interaction between YTHDF2 proteins and m6A-mRNAs (~2.54uM) (Zhu et al., (2014)). (2) BoxB containing RNAs do not have to compete with other mRNAs for binding to lambdaN-fusion proteins in the tethering assay. In contrast, m6A containing RNAs are likely competing with other m6A RNAs for binding to YTHDF proteins. (3) Overexpression of YTHDF proteins in the tethering assay leads to more stress granules forming in transfected cells which enhances general RNA recruitment. Therefore, this assay has many significant drawbacks that will likely artificially enhance the role that m6A can facilitate RNAs recruitment to stress granules.

In order to clarify how mRNAs are targeted to stress granules, we have added a discussion of how YTHDF proteins and G3BP proteins are similar in that they can be required for stress granule formation but do not affect the partitioning of individual mRNAs into stress granules (Figure 1 and (Matheny et al., 2021, RNA)). The resolution of this apparent conflict is due to the difference between examining the partitioning of an individual mRNA into stress granules as compared to examining the formation of stress granules in bulk. The targeting of any individual mRNA to a stress granule is a summation of many interactions and therefore any individual interaction makes little, or no, difference. In contrast, the overall assembly of a stress granule is a highly cooperative process and if the average interaction between many mRNAs is reduced even 5%, the assembly curve can shift into the regime where stress granules are not formed. We believe this additional discussion should help the reader understand how the results in this manuscript fit into the larger understanding of stress granule assembly.

Fig 3D: It is unclear how many repeats of experiments were performed.

We have now clarified this in the text.

Overall, the data presented suggests that the effect of m6A likely accounts for 15% to 20% as compared to the effect of mRNA length. The effect is there but not large in cell lines the authors studied. I am curious about HeLa cells.

We do acknowledge that m6A may, under some circumstances, make a small effect to the partitioning of mRNAs into stress granules. However, we emphasize that a direct test of m6A targeting mRNAs to stress granules failed to show any evidence for m6A targeting mRNAs to stress granules.

Reviewer #2 (Remarks to the Author):

In the manuscript “Limited effects of m6A modification on mRNA partitioning into stress granules”, Khong et al. analyze the effect of m6A modification on the recruitment of mRNAs into stress granules. Two studies (Anders et al, 2018; Ries et al, 2019) showed that m6A RNA modification has a role in the recruitment of RNAs to SGs through interaction with YTHDF proteins. Specifically, Anders et al showed the recruitment to be YTHDF3 dependent through stress-induced m6A in the 5' vicinity of the CDS. Here, the authors use smRNA FISH targeting Fem1b and Figl1 mRNAs on mES WT cells and methylation writer Mettl3 KO cells and find no differences in their recruitment to SGs. Linear regression analysis using annotated m6A sites (Xiang et al 2017) and RNAs recruited to SGs (Khong et al 2017) shows that overall m6A contributes slightly to SG recruitment. YTHDF1 and 2 tethering assays reveal an increase in the recruitment to SGs. Overall, using a combination of these approaches, the authors find that m6A modification and YTHDF binding play a role in stress granule recruitment but a minor one, which contradicts the findings of the two previous studies.

The topic of what RNA features affect their recruitment to stress granules has been a subject of extensive studies and it is of great interest to the community. The experimental results presented here are thought provoking. However, it is important to note that although the authors refer to two previous studies (Anders et al, 2018; Ries et al, 2019), the model proposed by Anders et al is not addressed: i) the location of stress-induced m6A is not considered and ii) YTHDF3 contribution to SG recruitment is not addressed. My overall recommendation is to further develop the study to strengthen the conclusions. A more robust analysis of the relationship between m6A and SG recruitment is needed. However, I am afraid that unless the conclusions are different from previous studies, the work might in general lead to repetition of previous works.

Major comments:

1. The two previous studies show that m6A modification enhances or facilitates the recruitment of mRNA to stress granules, while Khong et al show that m6A modification has a limited effect. Quantitatively speaking, it is difficult to discern enhancing, facilitating or limiting. I would suggest that the comparison is done more quantitatively and clearly.

Wherever possible, we have clarified our statements with quantitative numbers.

2. In general, both studies (Anders et al, 2018; Ries et al, 2019) are referred to as showing that overall m6A aids in the recruitment to SGs through interaction with YTHDF proteins. Nevertheless, these two studies are conceptually different and propose different models of m6A contribution to SG recruitment, and the authors address only the Ries et al model.

We have now looked at two RNAs with 5'UTR m6A modification. The mRNAs do not change in stress granule localization between WT and METTL3 KO mES cells. Since we only looked at two RNAs, we have also altered the manuscript to discuss possible conditions where m6A modification might affect stress granule partitioning of mRNAs.

3. The original study by Batista et al found that Mettl3 KO results in ~60% reduction of m6A in Mettl3 KO mESC. SmRNA FISH convincingly shows that both RNAs are recruited to SGs in Mettl3 KO, but supporting evidence is needed. Is the methylation of Fem1b and Figl1 RNAs reduced? Is there a gain of methylation upon stress? Please complement the smRNA FISH experiments by analysis of Fem1b and Figl1 methylation using RIP or a similar approach.

The reviewer points out that we do not know how much the methylation of the Fem1b and Figl1 mRNAs is reduced in the METTL3 KO mES cells. This is a valid point and to address this issue we have examined 11 additional mRNAs known to have reduced methylation in METTL3 KO mES cells compared to wildtype mES cells (Batista et al., (2014)). The stress granule localization of all 11 of these m⁶A mRNAs are not statistically different between wildtype and METTL3 KO mES cells. In fact, unexpectedly, 9 out of 11 mRNAs show slightly increased stress granule localization in METTL3 KO mES cells. This reinforces our conclusion that m6A methylation does not alter the localization of mRNAs into stress granules.

4. Several questions remain regarding the two genes chosen for smRNA FISH. It is not clear why Fem1b and Figl1 mRNAs were chosen, apart from their use by Ries et al, and the choice does not seem drawn in a representative manner. Are Fem1b and Figl1 methylated in mES cells where RNA FISH was performed? Since the conclusions are extrapolated to a global scale, I would suggest increasing the pool of RNAs to those that are a representative subset (methylated, interact with YTHDF proteins, recruited to SGs and analyze their recruitment in the absence of methylation). In making this choice, the authors should bear in mind the location of m6A sites, induction of m6A by stress and YTHDF1, 2 or 3 binding.

As described above, we have improved the manuscript by looking at additional mRNAs known to have reduced methylation in METTL3Δ cells. In addition, we also classified where the m6A mapped sites are on these mRNAs (shown in Figure S1B). While it is possible that different YTHDF proteins have different effects, current results suggest all three YTHDF proteins bind the same sites in mRNAs and exert the same functional consequences (Zaccara and Jaffrey, 2020, Cell).

5. It was shown that RNAs with stress-induced m6A around the 5' vicinity of their CDS (not overall methylation) are recruited to SGs by interacting with YTHDF3. Where do the m6A sites of these two mRNAs reside? The linear regression model and smRNA FISH assay do not consider the relative location of the m6A within the transcript. An additional caveat with the

linear regression analysis is that it uses datasets from different studies, thus the methylation status of RNA might be different. The authors should consider the pattern of m6A methylation in their analyses to be consistent with the model they are addressing.

In the comment, the reviewer suggests we be cognizant of the location of the m6A sites, and the possibility that different locations of modification have different outcomes. Since our analysis now includes mRNAs with different sites of m6A modifications, and none showed any differences in stress granule recruitment with changes in m6A modification, the simplest interpretation is m6A modification does not affect mRNA partitioning into stress granules. However, we can envision specific cases where it might matter, and we now raise those in the discussion.

6. Are YTHDF proteins recruited to SGs in mES WT and Mettl3 KO cells?

Ries et al., (2019) show YTHDF protein recruitment to stress granules is reduced in Mettl3 KO cells compared to WT mES cells.

7. YTHDF tethering assay is performed to determine whether YTHDF proteins have a role in the recruitment of mRNA to SGs. I have two concerns about the assay. First, there are a total of 25 binding sites. As authors note, this is beyond the average number of m6A sites in endogenous RNAs, which leads them to conclude that the observed positive effect is due to a larger number of binding sites and likely much lower for endogenous RNAs. This raises the question of why 25 binding sites in the first place? Why not a number closer to biological levels or to determine the linearity of the effect? In order to place the conclusions in a biological context, the number of binding sites in the tethering assay should be closer to what authors are trying to address.

In this comment the reviewer is suggesting we look at the effects of tethering YTHDF proteins with a smaller number of boxB sites. We have not performed this experiment for three reasons. First, the critical experiment of testing whether m6A modification affects stress granule recruitment of mRNAs in a METTL3 Δ cell line has shown no effect of m6A modification. Second, the tethering assay may have significant drawbacks that would likely overemphasize the role of recruiting m6A RNA to stress granules. (1) First, the Kd for BoxB elements binding to LambdaN proteins are in the order of ~ 1.3 nM (Keryer-Bibens et al., (2008)). This is approximately 500-fold greater than the interaction between YTHDF2 proteins and m6A-mRNAs (~ 2.54 μ M) (Zhu et al., (2014)). (2) BoxB containing RNAs do not have to compete with other mRNAs for binding to lambdaN-fusion proteins in the tethering assay. In contrast, m6A containing RNAs are likely competing with other m6A RNAs for binding to YTHDF proteins. (3) Overexpression of YTHDF proteins in the tethering assay leads to more stress granules forming in transfected cells which enhances general RNA recruitment. We did not examine cells with YTHDF proteins over-expressed to the level where they form constitutive stress granules but are concerned, even at lower expression levels, this might affect the analyses. Therefore, this assay has many significant drawbacks that will likely artificially enhance the role that m6A can facilitate RNAs recruitment to stress granules. Finally, we think examining a large number of tethering sites reveals the maximum possible effect m6A could have, and by analogy to tethering G3BP1 (Matheny et al., 2021, RNA), would be expected to drop with fewer sites.

We have maintained the tethering experiment in the manuscript for completeness and to show the results of all possible experiments. However, if the reviewers and editor prefer, we remove this experiment from the manuscript because of its limitations, we would be willing to do so.

Secondly, the authors use YTHDF 1 and 2 binding assay, while Anders et al propose a model wherein mRNAs associate with SGs by stress induced methylation in a YTHDF3-dependent manner. Considering that they are directly addressing that study, I would suggest the expansion of tethering experiments to include YTHDF3.

We note that a recent study suggests all YTHDF proteins are redundant in mES cells (Zaccara and Jaffrey (2020)). We also want to note that Anders et al., (2018) show a very subtle increase in m⁶A levels on mRNAs during cellular stress. Moreover, there was no quantification shown for the one RNA that changes in localization when they removed DRACH sequence.

8. The direct evidence on a global scale that would show that m⁶A does not play a role in SG recruitment is to analyze the RNA content and RNA methylome of SGs in WT and KO cells.

While we agree with the reviewer a transcriptome analysis would be useful, we have not pursued this approach because the reagents required to grow sufficient mES for stress granule purification and analysis would have been extremely expensive. Instead, we examined 11 additional mRNAs by smFISH with documented changes in m⁶A levels between WT and METTL3 KO mES cells. We see no difference in localization of these mRNAs to stress granules in METTL3 Δ cells. Given this, we feel confident that m⁶A modification does not generally increase mRNA partitioning into stress granules.

Minor comments:

1. Lines 26, 60 and similar: please mention the two mRNAs for which this was shown by smRNA FISH; the current formatting can be easily understood as all m⁶A-modified RNAs.

We have clarified the text wherever possible to be explicit about what results are seen with specific mRNAs, and where we make extrapolations to wider groups of mRNAs.

2. Fig 1B and Fig 3 – Please note what each data point represents in the figure. It is important to note how many individual cells and stress granules were quantified because the total number of FISH spots is not informative enough (we do not know how many FISH spots one cell has). One biological replicate should have a representative set of cells/SGs quantified.

We have now noted this in the manuscript.

3. Fig 3C – how many cells and biological replicates?

We have now noted this in the manuscript.

4. Tethering assay. When showing the relative portion of RNA in SGs per cell, the representative

images would be better interpretable if the image would highlight the boundaries of one cell. Is the amount of luciferase RNA constant between different samples?

The amount of luciferase RNA is relatively similar between individual cells since this is a clonal cell line. The images are zoomed in greatly such that individual fish spots and stress granules can be easily seen by readers. Cells often extend beyond what the image show.

5. As m6A modification is implicated in regulating RNA metabolism, was the total number/stability of Fem1b and Figl1 mRNAs affected by the loss of methylation?

We saw no statistical difference in the number of Fem1b or Figl1 mRNAs in the METTL3Δ as compared to wild type cell line. Since we see no difference, and this does not influence the conclusions in the manuscript, we have not included this data in the revision. However, if this was deemed to be important, we would be happy to add it in. It is also worth noting that recent results suggest that the effect of m6A modification on mRNA levels is quite small (on average ~20%, (Zaccara and Jaffrey, Cell (2020)), which suggests we should not generally see significant differences in individual mRNA levels in the two cell lines.

6. The results in Fig. 3D, as authors note in the manuscript, show that YTHDF1 and 2 tethering has an effect on recruitment to SGs. Thus, it is puzzling that this effect is disregarded for the number of used binding sites, without knowing the linearity of the effect.

In our previous work, we have determined how the recruitment to stress granules is affected by the number of boxB sites for TIA1 and G3BP1 (Matheny et al., 2021 (RNA)). If we extrapolate from these experiments to the results from the YTHDF tethering experiments, we can infer that tethering 4 YTHDF proteins would make a very minor change to mRNA partitioning into stress granules (predict from 11% to 12-13%). We have not added this cross comparison to the manuscript since a) we are cognizant of the limitations of the tethering experiments (discussed above) and b) it requires inferring the results are similar for YTHDF to TIA1 and G3BP1, which although likely to be true requires an assumption that may be unfounded.

7. Line 181 – please note this is the average±

We have now included this in the manuscript.

8. Line 189, 191 – “small effect”, “small role”, please use a quantitative description of the data, while the interpretation if this is a small or physiologically relevant contribution in the discussion.

We have included more quantitative description of the data.

9. Line 344 – Figure 3 not 2

We thank the reviewer for catching this mistake.

Reviewer #3 (Remarks to the Author):

Stress granules are cytoplasmic membrane-less condensates that are formed in diverse stress conditions. The condensate is comprised of non-translating mRNPs. Although the protein components of stress granules were around, the mRNA content began to be revealed only recently (Khong et al., 2017; Namkoong et al., 2018). A major conclusion from these studies is that longer mRNAs tend to localize more in the stress granule. Despite the length difference, mRNAs of the similar length show different recruitment into stress granules, raising a possibility of sequence-dependent targeting of mRNA transcripts. Indeed, recent works (Anders et al., 2018; Ries et al., 2019) showed that m6A modifications promote mRNA recruitment to stress granules through the interaction with YTHDF proteins. In this paper, Khong et al. argues, based on three major reasonings listed below, that m6A modifications are unlikely to play a significant role in recruiting mRNAs to stress granules.

First, the authors examined the localization of two poly-m6A mRNAs, FIGNL1 and FEM1B, in wildtype and Δ METTL3 mES cells in stress granules and found that these two mRNAs are enriched similarly in stress granules in both cell types. These particular mRNAs are examples of polymethylated mRNA and of showing higher smFISH localization signals in the stress granule compared to non-methylated mRNA (Ries et al., 2019). Since METTL3 is a component of m6A methylase, based on their observation, the authors claimed that m6A does not play a major role in the localization of mRNAs into stress granules.

Second, the authors performed multiple regression analyses where they compared the effect of mRNA length on stress granule targeting with or without an additional contribution of m6A modifications. Their analyses showed that the ORF length is a much stronger predictor for mRNA partitioning than m6A modifications, and for fixed transcript lengths, m6A modifications up to 4 m6A lead to an increase in predicted stress granule enrichment from ~ 15% (0 m6A) to ~ 22% (4 m6A).

Third, they used an artificial tethering assay based on the λ N-BoxB system to ask if the tethering of YTHDF proteins on the reporter mRNA leads to recruitment of the mRNA into the stress granule. Based on the change in the recruitment from 11% to ~ 20%, they claimed that most endogenous mRNAs with smaller number of m6A sites than the reporter mRNA are unlikely to be affected by m6A modifications in the recruitment to stress granules.

I think these reasonings are poorly supported as I discuss in detail below. I suggest that the authors focus on providing extra transcriptome-level localization data in Δ METTL3 mES cells since other two datasets (regression analysis and artificial tethering assay) are only indirect evidence and subject to different interpretations.

Major concerns

1. The smFISH experiment in Δ METTL3 mES cells: In this experiment, their conclusion is drawn from the localization pattern of just two mRNAs, FIGNL1 and FEM1B, which shows a minimal change upon the knock-out of METTL3. In Fig 1A, they reported that the amount of m6A is reduced overall by ~ 70% in Δ METTL3 mES cells. However, the degree of reduction in m6A modification in the mRNAs being tested is unknown. Without a detailed analysis on m6A levels for these transcripts, it is hard to make a conclusion regarding the effects of m6A modifications on the localization of particular transcripts. In addition, the number of mRNAs tested, 2 here, is highly limited. In fact, previous works (Fig 4B in Ries et al., 2019 and Fig 1B in Fu & Zhuang, 2020) already showed that the m6A level in a given mRNA is not a determining

factor for stress granule recruitment but provides only an overall tendency for the recruitment: For a given number of m⁶A, stress granule enrichments are broadly distributed with significant overlaps between neighboring populations. I thus suggest that the authors examine the localization change in a much larger group of mRNAs. The transcriptome-level analysis would provide unambiguous evidence to test their claim.

In this comment the reviewer makes several related points that we address in turn. First, he/she argues that we need to a) know how methylation of the mRNAs we examine is affected in METTL3Δ cells, and b) examine more mRNAs. We have addressed these concerns by examining 11 additional mRNAs that have previously been demonstrated to show reduced m⁶A levels in METTL3Δ cells (Batista et al., 2014). Since none of the mRNAs examined shows decreased stress granule partitioning, we are confident that m⁶A modification plays little or no role in recruiting mRNAs into stress granules.

In a second comment, the reviewer argues that Ries et al only " showed that the m⁶A level in a given mRNA is not a determining factor for stress granule recruitment but provides only an overall tendency for the recruitment", which implies that our demonstration that m⁶A has essentially no effect is not important. We disagree with this reviewer's assessment of the previous work. In fact, Ries et al. concluded "**Our studies demonstrate that m⁶A regulates the fate of cytosolic mRNA by scaffolding DF proteins, which leads to the formation of phase-separated DF-m⁶A-mRNA complexes that then partition into phase-separated structures in cells.**" Moreover, many published papers that are citing Ries et al., (2019) including Jaffrey's group further state the conclusion that YTHDF proteins recruit m⁶A RNA to stress granules. We provide a subset of quotes from different papers citing Ries et al., (2019)).

"For example, in cell stress, DF proteins bind m⁶A-mRNA and relocalize to stress granules, but the mRNA is not targeted for degradation (Ries et al., (2019))" (Zaccara and Jaffrey (2020) Cell)

"Previous studies have found that YTHDFs direct m⁶A-mRNAs to through condensates mediated by liquid-liquid phase separation (LLPS) in the cytoplasm (Cheng et al., (2021) Cancer Cell)

"m⁶A-modified RNAs have an enhanced ability to phase separate, and this tracks positively with a report that m⁶A-modified RNAs are recruited to stress granules by interacting with YTHDF family of proteins (Ries et al., (2019))" (Sidibe et al., (2020) Journal of Neurochemistry)

"m⁶A has also been shown to enhance phase separation of YTHDF m⁶A reader proteins and regulate their subcellular localization to phase-separated RNP granules, such as stress granules (Ries *et al.*," (He and He (2021) EMBO)

Given the widespread, yet erroneous, conclusion that m⁶A modification through the binding of YTHDF proteins targets mRNA to stress granules, we believe our manuscript is important to correct this issue.

2. Multiple regression analyses: In Fig 2E and 2G, the regression model predicted that the fraction of 2.5kb RNA within stress granules increases from 15% (0 m⁶A) to 22% (4 m⁶A) as the m⁶A modification varies. However, the way data is present is misleading. When a given

mRNA without any m6A is modified to have 4 m6A sites, the predicted change of the mRNA fraction in the stress granule is in fact ~ 50% increase ($= (22-15)/15$) from the original value. It should be noted that the overall fraction of bulk mRNA within the stress granule is very low (~ 10%) (Khong et al., 2017; Namkoong et al., 2018). Moreover, the length of mRNA is not a major target of regulation once the transcript is formed. From the perspective of localization regulation, the predicted change may not be a small amount. I suggest that rather than the model prediction of which result may be differently interpretable, the authors focus on providing extra experimental evidence to support their claims.

We agree with the reviewer that “providing extra experimental evidence” is the key. Thus, we examined 11 additional RNAs with m6A binding sites. We see no difference in the stress granule localization between wildtype and METTL3 KO mES cells.

3. BoxB- λ N reporter assay: The data seem solid but regarding interpretation I still find the same issues as above. The authors indeed observe an increase in the stress granule partitioning of the reporter mRNA from 11% to 21% when tethered by YTHDF. This corresponds to almost 100% increase compared to the untethered case.

In this comment the reviewer raises the issue of how one should best interpret the effects of tethering 25 YTHDF proteins to a reporter mRNA. In order to keep an open mind, we reported this experiment to raise the possibility that YTHDF proteins might affect mRNA recruitment to a small degree. However, we believe this experiment is limited for the following reasons. First, the tethering assay may have significant drawbacks that would likely overemphasize the role of recruiting m6A RNA to stress granules. (1) The K_d for BoxB elements binding to LambdaN proteins are in the order of ~1.3nM (Keryer-Bibens et al., (2008)). This is approximately 500-fold greater than the interaction between YTHDF2 proteins and m6A-mRNAs (~2.54uM) (Zhu et al., (2014)). (2) BoxB containing RNAs do not have to compete with other mRNAs for binding to lambdaN-fusion proteins in the tethering assay. In contrast, m6A containing RNAs are likely competing with other m6A RNAs for binding to YTHDF proteins. (3) Overexpression of YTHDF proteins in the tethering assay leads to more stress granules forming in transfected cells which enhances general RNA recruitment. Therefore, this assay has many significant drawbacks that will likely artificially enhance the role that m6A can facilitate RNAs recruitment to stress granules. Given this, we interpret the result to show a minimal effect, at most, of YTHDF proteins, but report the result in fairness.

Minor points

1. (In Fig 1C legend) Are these data from U2OS cells? If they are not typos, I suggest they repeat this analysis for mES cells for consistency. Also, the number of images/cells analyzed must be specified.

We have fixed the legend.

2. (Fig 1B) PABP -> G3BP?

We have fixed the figure.

3. (page 5, line 89-91) This sentence needs to be modified.

We have modified the sentence for clarity.

REVIEWER COMMENTS

Reviewer #1 (Remarks to the Author):

The authors have addressed most of my comments. I still think there are two points the authors will need to consider.

mESCs are very different from cancer cells. m6A methylation behaves quite differently in different cellular contexts, even different cancer cell lines. The type of transcripts methylated and relative levels of YTHDF proteins and their binding partner proteins can exist in different levels with different PTM. I strongly urge the authors to clearly limit this study to mESCs. This example does suggest that in certain cells (or most primary cells or tissues) m6A may not play a big role in stress granule localization, definitely not mESCs. In other cell types, including neurons, granule dynamics (may not be stress granule) is a critical factor of m6A biology.

The data suggest to me the effect is limited but again the authors may not be able to completely exclude stress granule localization of all m6A-modified transcripts. "Limited effect" is appropriate.

Reviewer #2 (Remarks to the Author):

The authors have partially addressed all the comments from the previous round of review, which has improved the manuscript. I would encourage a direct assessment of RNA methylation of these transcripts in stress granules in the WT and KO. This information seems crucial since this is the key point the authors are addressing.

The issues with the YTHD tethering assay remain. The authors note approximately a 2-fold increase in the recruitment to stress granules, thus not very small. The authors recognize the difficulty to correlate the tethering assay with the endogenous RNA behavior, hence the message of this experiment remains open to interpretation.

Reviewer #3 (Remarks to the Author):

I feel that the revised manuscript is indeed improved, and it is good to see that efforts have been made to provide extra smFISH data. However, I still feel that this manuscript is not ready for publication in Nature Communications for reasons I discuss in details below:

1. The key message the authors try to deliver in the manuscript is the minimal role of m6A modifications on mRNA partitioning into stress granules. To support this argument, in the revised manuscript, the authors have included smFISH data for 11 additional mRNAs, measured in WT and METTL3 KO mES cells (Fig. 1). They found no difference in the fraction of mRNAs in stress granules between WT and METTL3 KO cells (concerns regarding technical issues are discussed below), which leads the authors to conclude

that m6A modifications do not alter mRNA partitioning into stress granules.

The critical element in this argument is whether and how much reduction occurs in methylation of those specific mRNA species tested upon METTL3 KO. For this, although not specifically mentioned in Fig. S1B legend or methods, the authors seem to use “documented changes in m6A levels” from Batista et al (2014). I suggest the authors independently confirm the changes in the m6A levels for those mRNAs used in this work using gene-specific techniques such as RIP-qPCR. This is particularly important considering the possibility of stress-induced differential changes in the m6A levels. In the point-by-point response, the authors mentioned “Anders et al (2018) show a very subtle increase in m6A levels on mRNAs during cellular stress”. However, the degree of m6A changes during stress may vary depending on specific mRNA species. In short, the authors do not provide the exact status of m6A levels for mRNAs used to produce their smFISH data.

2. It is known that the overall fraction of mRNAs within stress granules is typically quite low (10-20%, Khong et al., 2017, Namkoong et al., 2018), as also shown in Fig 2B-D. However, the fractions of those mRNAs in Fig 1 and S1 are all very high: most of them are > 40%, some are > 60%. Does this mean these transcripts are rather special ones partitioning very strongly into stress granules? Is it due to the fact that they contain multiple m6A modifications? Isn't it possible that the effect of m6A on stress granule partitioning is highly nonlinear and even with overall reduction in m6A modifications, they still have enough modifications to partition well into stress granules? This last point also raises an issue for interpreting the result of YTHDF tethering experiments.

3. Regarding the YTHDF tethering experiments, the underlying assumption is that tethering more YTHDF to mRNAs would lead to a monotonic increase in stress granule partitioning. However, the authors acknowledge that this “assumption may be unfounded”. In addition, with 25 GFP labeled, individual luciferase mRNA should be identifiable as GFP puncta (similar approaches have been widely used for direct RNA imaging). However, in Fig. 3C, red “tethered protein” channels do not show any individual puncta outside of stress granules while many white puncta, corresponding to individual luciferase RNA examined by smFISH, are present in the MERGE channel. Does this suggest that YTHDF tethering experiments are performed somehow in the non-saturating condition of YTHDF/G3BP1-GFP-λN (i.e. not all BoxB sites are bound by GFP fusion)?

4. (smFISH quantification) In many smFISH data (Fig 1), red puncta, corresponding to individual mRNAs, often overlap one another (for example, Bahcc1 in METTL3 KO cells and Aff1 in WT). How are these overlapping puncta processed in quantification? Also, are the mRNA fractions in stress granules computed with respect to all mRNAs including nuclear ones? If that is the case, isn't it more appropriate to consider only those in cytoplasm?

5. Page 6, line 117: “Figure 1C-D”. Currently, there are no subfigures in Fig 1.

6. Page 12, line 261: “there could mRNAs where”

7. Page 23, line 484-488: no corresponding figure for “(C) Scatter plot of the fraction of RNA molecules in stress granules in wildtype and 485 Δ METTL3 mES cells”

REBUTTAL LETTER

Reviewer #1 (Remarks to the Author):

The authors have addressed most of my comments. I still think there are two points the authors will need to consider.

mESCs are very different from cancer cells. m⁶A methylation behaves quite differently in different cellular contexts, even different cancer cell lines. The type of transcripts methylated and relative levels of YTHDF proteins and their binding partner proteins can exist in different levels with different PTM. I strongly urge the authors to clearly limit this study to mESCs. This example does suggest that in certain cells (or most primary cells or tissues) m⁶A may not play a big role in stress granule localization, definitely not mESCs. In other cell types, including neurons, granule dynamics (may not be stress granule) is a critical factor of m⁶A biology.

The data suggest to me the effect is limited but again the authors may not be able to completely exclude stress granule localization of all m⁶A-modified transcripts. "Limited effect" is appropriate.

We agree that mESCs are very different from cancer cells and there is a possibility that RNA methylation may behave differently in different cellular contexts. However, we also want to note that we did look at U-2 OS cells by two methodologies. (1) First, we asked if there is a correlation between m⁶A RNAs and stress granule enrichment in U-2 OS cells datasets when length is normalized. We see at best a very weak correlation. (2) Second, we tethered YTHDF proteins on boxB luciferase RNAs and at best, with 25 binding sites, we can only get a small fraction of the boxB luciferase RNAs in stress granules in U-2 OS cells. Nevertheless, we agree with the reviewer that m⁶A could provide some effect albeit limited, and we have written the manuscript to acknowledge this possibility.

Reviewer #2 (Remarks to the Author):

The authors have partially addressed all the comments from the previous round of review, which has improved the manuscript. I would encourage a direct assessment of RNA methylation of these transcripts in stress granules in the WT and KO. This information seems crucial since this is the key point the authors are addressing.

The issues with the YTHD tethering assay remain. The authors note approximately a 2-fold increase in the recruitment to stress granules, thus not very small. The authors recognize the difficulty to correlate the tethering assay with the endogenous RNA behavior, hence the message of this experiment remains open to interpretation.

We agree with the reviewer that a direct assessment of RNA methylation of the transcripts in stress granules will provide increased confidence in the data. We have now done that. We looked at 10 different m⁶A sites that were shown to be depleted by Batista et al., (2014) and confirmed that it is indeed depleted of m⁶A (~50-60%) by m⁶A-IP in our hands in the metl13Δ cells we are using. In communication with Batista and examining his raw data, we learned that

Fem1b, and Figl1 RNAs were not depleted of m⁶A by m⁶A mapping, which we reproduced in our m⁶A-IP qRT-PCR. Finally, consistent with the behavior of m⁶A (promote RNA decay), we see METTL3 KO cells result in an increase in RNA levels by qRT-PCR.

We agree with the reviewer that there is a 2-fold increase, but we note that the fraction of luciferase molecules is minor (~20%) even with 25 binding sites for YTHDF proteins. This indicates m⁶A is not a dominant factor that can recruit the majority of RNAs into stress granules and argues that many other RNA-Bps and RNAs are possibly involved in the process.

Reviewer #3 (Remarks to the Author):

I feel that the revised manuscript is indeed improved, and it is good to see that efforts have been made to provide extra smFISH data. However, I still feel that this manuscript is not ready for publication in Nature Communications for reasons I discuss in details below:

1. The key message the authors try to deliver in the manuscript is the minimal role of m⁶A modifications on mRNA partitioning into stress granules. To support this argument, in the revised manuscript, the authors have included smFISH data for 11 additional mRNAs, measured in WT and METTL3 KO mES cells (Fig. 1). They found no difference in the fraction of mRNAs in stress granules between WT and METTL3 KO cells (concerns regarding technical issues are discussed below), which leads the authors to conclude that m⁶A modifications do not alter mRNA partitioning into stress granules.

The critical element in this argument is whether and how much reduction occurs in methylation of those specific mRNA species tested upon METTL3 KO. For this, although not specifically mentioned in Fig. S1B legend or methods, the authors seem to use “documented changes in m⁶A levels” from Batista et al (2014). I suggest the authors independently confirm the changes in the m⁶A levels for those mRNAs used in this work using gene-specific techniques such as RIP-qPCR. This is particularly important considering the possibility of stress-induced differential changes in the m⁶A levels. In the point-by-point response, the authors mentioned “Anders et al (2018) show a very subtle increase in m⁶A levels on mRNAs during cellular stress”. However, the degree of m⁶A changes during stress may vary depending on specific mRNA species. In short, the authors do not provide the exact status of m⁶A levels for mRNAs used to produce their smFISH data.

We like to thank the reviewer for the comment. We now looked at multiple m⁶A sites by m⁶A-IP qRT-PCR in the mRNAs we examined by smFISH and demonstrate that they show reduced methylation in the METTL3 KO cells (essentially reproducing the results of Batista et al (2014)).

We also looked at whether m⁶A level changes during arsenite stress and we did not see any changes of m⁶A levels on total mRNAs during stress in either WT or METTL3 KO cells (WT levels are shown in Figure on the right, two biological replicates). This argues that there are no substantial changes in m⁶A levels during stress. Moreover, we note that Ries et al., (2019) used unstressed m⁶A mapping data set to make their conclusions that m⁶A plays a dominant role in recruiting RNA to stress granules.

2. It is known that the overall fraction of mRNAs within stress granules is typically quite low (10-20%, Khong et al., 2017, Namkoong et al., 2018), as also shown in Fig 2B-D. However, the fractions of those mRNAs in Fig 1 and S1 are all very high: most of them are > 40%, some are > 60%. Does this mean these transcripts are rather special ones partitioning very strongly into stress granules? Is it due to the fact that they contain multiple m⁶A modifications? Isn't it possible that the effect of m⁶A on stress granule partitioning is highly nonlinear and even with overall reduction in m⁶A modifications, they still have enough modifications to partition well into stress granules? This last point also raises an issue for interpreting the result of YTHDF tethering experiments.

The reason why these RNAs partitioning really highly is because almost all RNAs with multiple m⁶A sites are long mRNAs. And length is a strong metric for RNA localization to stress granules.

There is a possibility that the m⁶A contribution is nonlinear. However, we want to note that at best with luciferase RNA, we get 20% of the RNA in stress granules with 25 YTHDF binding sites (10% increase compared to 0 YTHDF binding sites). This suggests m⁶A cannot be the only explanation and that other factors must play a role in recruiting RNA to stress granules. If the first m⁶A site contributes half of that 10%, that's only a 5% increase. This suggests the RNAs that accumulate in stress granules with only 6 m⁶A sites and 60% partitioning cannot solely be explained by those six m⁶A sites. There must be additional elements besides m⁶A that contribute to the partitioning of the RNA to stress granules to explain the additional 40% variance (additional RNA-Binding protein sites etc).

3. Regarding the YTHDF tethering experiments, the underlying assumption is that tethering more YTHDF to mRNAs would lead to a monotonic increase in stress granule partitioning. However, the authors acknowledge that this "assumption may be unfounded". In addition, with 25 GFP labeled, individual luciferase mRNA should be identifiable as GFP puncta (similar approaches have been widely used for direct RNA imaging). However, in Fig. 3C, red "tethered protein" channels do not show any individual puncta outside of stress granules while many white puncta, corresponding to individual luciferase RNA examined by smFISH, are present in the MERGE channel. Does this suggest that YTHDF tethering experiments are performed somehow in the non-saturating condition of YTHDF/G3BP1-GFP-λN (i.e. not all BoxB sites are bound by GFP fusion)?

We want to note that in order to see single-molecule RNA using systems like boxB or MCP, researchers typically do other manipulations to enhance the signal. For example, it is typical that researchers put a NLS site on the fusion protein in order to limit the signal produced from overexpression of the tethered binding protein in the cytoplasm (Latallo et al., (2019) Methods). So, we think this is the reason why we do not see GFP punctas. There are no introduced NLS sites for the G3BP or YTHDF tethered proteins. It is still possible that we don't have every boxB binding site interacting with YTHDF proteins, but we want to note that is why we engineer 25 sites. Most mRNAs have less than 5 m⁶A binding sites.

4. (smFISH quantification) In many smFISH data (Fig 1), red puncta, corresponding to individual mRNAs, often overlap one another (for example, Bahcc1 in METTL3 KO cells and Aff1 in WT). How are these overlapping puncta processed in quantification? Also, are the mRNA fractions in stress granules computed with respect to all mRNAs including nuclear ones? If that is the case, isn't it more appropriate to consider only those in cytoplasm?

There is a lot of overlap when we z-stack our images as shown in the manuscript. But we quantify our smFISH images in 3D using Imaris software. There are very few overlapping puncta in 3D (Movies attached to the manuscript). The mRNAs computed only include cytoplasmic RNAs. We used Imaris to mark where the nucleus is, the cytoplasm, and whether they are in stress granule to distinguish smFISH spots localization. We have now noted this in the manuscript. Thank you for asking for clarification.

5. Page 6, line 117: "Figure 1C-D". Currently, there are no subfigures in Fig 1.

6. Page 12, line 261: "there could mRNAs where"

7. Page 23, line 484-488: no corresponding figure for "(C) Scatter plot of the fraction of RNA molecules in stress granules in wildtype and 485 Δ METTL3 mES cells"

We thank the reviewer for the careful reading and for the comments 5-7 and we now have adjusted the manuscript accordingly.